# Investigating the Differential Circulating microRNA Expression in Adolescent Females with Severe Idiopathic Scoliosis: A Proof-of-Concept Observational Clinical Study

**DOI:** 10.3390/ijms25010570

**Published:** 2024-01-01

**Authors:** Lavinia Raimondi, Angela De Luca, Alessia Gallo, Fabrizio Perna, Nicola Cuscino, Aurora Cordaro, Viviana Costa, Daniele Bellavia, Cesare Faldini, Simone Dario Scilabra, Gianluca Giavaresi, Angelo Toscano

**Affiliations:** 1Scienze e Tecnologie Chirurgiche, IRCCS Istituto Ortopedico Rizzoli, Via di Barbiano, 1/10, 40136 Bologna, Italy; lavinia.raimondi@ior.it (L.R.);; 2Dipartimento di Ricerca, IRCCS ISMETT (Istituto Mediterraneo per i Trapianti e Terapie ad Alta Specializzazione), 90127 Palermo, Italy; 3Ortopedia Generale, IRCCS Istituto Ortopedico Rizzoli, Via di Barbiano, 1/10, 40136 Bologna, Italyangelo.toscano@ior.it (A.T.); 4Clinica Ortopedica e Traumatologica I, IRCCS Istituto Ortopedico Rizzoli, Via di Barbiano, 1/10, 40136 Bologna, Italy; 5Fondazione Ri.MED, Dipartimento di Ricerca IRCCS ISMETT, Via Ernesto Tricomi 5, 90145 Palermo, Italy

**Keywords:** adolescent idiopathic scoliosis, microRNAs, osteogenesis

## Abstract

Adolescent Idiopathic Scoliosis (AIS) is the most common form of three-dimensional spinal disorder in adolescents between the ages of 10 and 18 years of age, most commonly diagnosed in young women when severe disease occurs. Patients with AIS are characterized by abnormal skeletal growth and reduced bone mineral density. The etiology of AIS is thought to be multifactorial, involving both environmental and genetic factors, but to date, it is still unknown. Therefore, it is crucial to further investigate the molecular pathogenesis of AIS and to identify biomarkers useful for predicting curve progression. In this perspective, the relative abundance of a panel of microRNAs (miRNAs) was analyzed in the plasma of 20 AIS patients and 10 healthy controls (HC). The data revealed a significant group of circulating miRNAs dysregulated in AIS patients compared to HC. Further bioinformatic analyses evidenced a more restricted expression of some miRNAs exclusively in severe AIS females. These include some members of the miR-30 family, which are considered promising regulators for treating bone diseases. We demonstrated circulating extracellular vesicles (EVs) from severe AIS females contained miR-30 family members and decreased the osteogenic differentiation of mesenchymal stem cells. Proteomic analysis of EVs highlighted the expression of proteins associated with orthopedic disease. This study provides preliminary evidence of a miRNAs signature potentially associated with severe female AIS and suggests the corresponding vesicular component may affect cellular mechanisms crucial in AIS, opening the scenario for in-depth studies on prognostic differences related to gender and grade.

## 1. Introduction

Scoliosis is a spinal deformity characterized by lateral curvature of the spine of 10° or more and the rotation of the vertebra [1,2,3]. To classify scoliosis, four main factors are considered, although there is considerable variation between patients: age, curve size and location, and causative factors. Consequentially, various types of scoliosis are typically reported: congenital, neuromuscular, degenerative, and idiopathic.

Adolescent scoliosis is defined as idiopathic (AIS) when the cause is unknown, accounting for around 80% of cases and affecting 0.5–3.0% of the paediatric population. About 0.3 to 0.5% of those affected exhibit a curvature greater than 20°, which generally necessitates treatment [4,5]. Several studies reported adolescent individuals had a higher prevalence of AIS than young children [6,7], highlighting puberty as a critical time for the development and worsening of AIS. Regarding gender, available data are sometimes controversial, with some studies showing a higher incidence of AIS in females, particularly after puberty, while others have not shown significant difference. It is notable small curvatures are equally prevalent in both girls and boys; however, higher Cobb angles are more common in girls, indicating severe forms of AIS primarily affect females after puberty [5,8].

An initial physical examination by orthopedists remains essential in the AIS diagnosis, both to exclude any other pathological conditions that may be responsible for the spinal deformity and to identify key indicators such as shoulder and scapula asymmetry, rib protrusion during forward flexion on the Adams test, and waist and trunk asymmetry [9]. The spinal X-ray remains the standard imaging method for the evaluation of AIS. The prediction of curve progression has been critically based on the assessment of skeletal maturity; indeed, it has been described the curvature progression was limited if the magnitude was less than 30 degrees, and the patient had reached skeletal maturity (a bone age of 15 years in girls and 17 years in boys). As for the treatment of AIS, surgical treatment is approved in patients with a still-immature skeleton and progressive scoliosis exceeding 45 degrees. Otherwise, bracing is routinely prescribed to patients with an immature skeleton and progressive scoliosis from 25 to 45 degrees [10,11,12]. 

The etiology of AIS is still debated, and the results shared are still very controversial. Many theories have been proposed, including genes and heredity, environment, hormones, metabolism, biochemistry, neurological factors, and asymmetric growth [13,14]. More recently, single-cell RNA sequencing approaches revealed differences in bone development-related cell differentiation between AIS patients and healthy control (HC). Moreover, a potential immunological pathogenesis of MSC differentiation dysfunction in AIS has been proposed [15], as well as a correlation among the aberrant DNA methylation and the formation of hemivertebra and congenital scoliosis [16,17].

Non-coding RNAs (ncRNAs) are taking on an emerging role in many orthopedic diseases [18,19,20,21], and among these also in scoliosis. In fact, a number of studies have shown there was a dysregulation of miRNAs in scoliosis and have linked this to the pathogenesis of the disease [22,23,24,25,26,27,28]. Based on their length, they have been classified into short non-coding RNAs (<200 nucleotides: microRNAs) or long non-coding RNAs (>200 nucleotides: lncRNAs and circRNAs). MicroRNAs (miRNAs) are regulatory molecules, affecting the stability and/or translation of target mRNAs. Aberrant regulation of various miRNAs has been reported in bone tissue during osteoporosis [29], in intervertebral disc degeneration pathogenesis [30],and osteoarthritis disease [31]. Interestingly, several miRNAs have been identified to be dysregulated in serum and plasma of patients with orthopedic diseases. Circulating miRNAs have been associated with bone mineral density [32] and proposed as diagnostic and prognostic biomarkers [33]. Overall, circulating miRNAs may provide useful diagnostic and therapeutic information because of their role in orthopedic disorders. They may reflect the disease state and are differentially affected by therapeutic treatment [34,35,36,37,38,39,40,41,42,43,44,45,46,47,48,49]. Small-RNA sequencing studies evidenced a series of circulating miRNAs differentially expressed in the AIS patients compared to the HC. Notably, some of these miRNAs targeted genes involved in cell signaling pathways that regulate the pluripotency of stem cells; others in osteoblast/osteoclast differentiation mechanism or in the mineralization of smooth muscle cells [22]. Overall, it has been proposed these miRNAs contributed to the epigenetic control of signaling pathways correlated with bone metabolism in healthy and patients. Dysregulation of circulating miRNAs in AIS may also affect osteocyte function, resulting in low bone mass. In this regard, Zang et al. also showed a significant negative correlation between circulating miR-145 levels and serum sclerostin, osteopontin, and osteoprotegerin in AIS [26]. As shown by Chen et al., the analysis of differentially expressed miRNAs in AIS bone and plasma samples represents a new source of biomarkers and actors in the aetiopathogenesis of AIS; for example, miRNA profiles in bone tissues from AIS and healthy controls identified the upregulation of miR-96-5p, which was further confirmed in plasma samples from AIS girls compared to HC [50].

As circulating miRNAs have been proposed as biomarkers for orthopedic disease, we aimed in this study to identify an aberrant circulating miRNA expression signature in AIS, able to distinguish patients from healthy controls. For this purpose, we analyzed the expression of a panel of circulating miRNAs from AIS and HC, through microfluidic cards. Data obtained revealed a peculiar expression of a small group of circulating miRNAs in the adolescent females with severe AIS; in particular, we found a dysregulation of circulating miRNAs belonging to the miR-30 family, described as miRNAs involved in bone metabolism [51,52]. We then investigated the effects of circulating extracellular vesicles (EVs) from severe AIS patients on human mesenchymal stem cells (hMSCs), during osteogenic differentiation. Osteogenic differentiation was impaired in severe AIS [53,54] and we asked if circulating EVs from plasma of severe AIS patients were involved in such phenomena. In fact, EVs can affect molecular mechanisms in bone turnover, in several bone disease [55,56] and can transfer a plethora of signaling molecules with the potential to modulate the phenotype of target cells [57,58,59]. We showed AIS-EVs expressed miR-30 family members, whose expression increased in hMSC treated with AIS-EVs. Moreover, AIS-EVs negatively affected osteogenic differentiation and impaired bone mineralization, probably also through their protein and miRNA content. These latter results led us to hypothesize the specific content of EVs (ncRNAs and proteins) may also contribute to the pathogenesis of AIS and should be further investigated.

## 2. Results

### 2.1. Clinical Features of AIS Patients

Twenty adolescent patients and ten age-matched healthy controls (HC) were included in the present study (Table 1). 

Enrolled patients were required to have a minimum of two years of follow-up of the pathology and no surgical treatment prior to enrollment in the study. The mean age of the patient group was 14.7 ± 1.5 years old, while the female-male ratio was respectively 5:1. The mean BMI value of this group was 21.6 ± 4.5. The Risser evaluation for patients was 3.4 ± 1.8. Ten HCs were enrolled, with a female-male ratio of 5:5 and a mean age of 15 ± 2.3. HC did not have to suffer from orthopedic or oncological diseases, nor to be close relatives of the patients. According to the Cobb angle, the distribution of all AIS patients was: sixteen patients with severe scoliosis, two patients with moderate scoliosis, and two patients with mild scoliosis. The three male patients were characterized by severe AIS. A more detailed description of every single patient is reported in Appendix A. The clinical protocol is described in detail in Section 4.

### 2.2. Identification of Circulating miRNAs Differentially Expressed in AIS Patients and Healthy Controls

The recent findings on biological characteristics of circulating miRNAs pointed out their value as potential biomarkers. To detect circulating miRNAs from plasma of AIS patients and HC, we used miRNA microfluidic array cards, analyzing a large panel of individual miRNAs assay. Data analysis showed a relevant group of circulating miRNAs were significantly and differentially expressed between AIS patients and HC (Appendix A). Among these, we found a small group of miRNAs, belonging to miR-30 family, strongly up-regulated in AIS patients. Several of these miRNAs were already known to be involved in bone metabolism and bone disease. In Figure 1, the volcano plot analysis showed the relationship between fold change and statistical significance of the differentially expressed circulating miRNAs, while the heat map diagram showed the cluster (Appendix A) and principal component analyses (Appendix A) of differentially expressed miRNAs. 

We then analyzed circulating miRNAs among females with severe AIS and HC females, thus excluding moderate females and severe males AIS. Notably, as shown in Figure 2, we found some miRNAs, such as hsa-miR-30d-5p, hsa-miR-30a-5p, hsa-miR-30a-3p, hsa-miR-30e-3p, exclusively expressed in females with severe AIS (Heat map and PCA analysis in Appendix A). 

Conversely, miR-30 family members were not significantly expressed in moderate AIS patients and in severe males AIS patients, as shown in the volcano plot analysis (Figure 3), heat map, and PCA analysis (Appendix A). 

In Figure 4, data are obtained from miRNet 2.0 software showed the miRNA-target interaction networks from analysis of AIS patients versus HC, while in Figure 5 are shown the miRNA-target interaction networks from analysis of severe female AIS versus females HC. 

Interestingly, the network analysis of circulating miRNA data followed by a functional enrichment analysis using a hypergeometric algorithm with various database (e.g., KEEG), indicated an association with several diseases, including scoliosis, spinal curvature, and other bone characteristics. We further investigated the genes involved in miRNA-target interaction networks by analyzing four miRNA types: hsa-miR-30a-5p, hsa-miR-30d-5p, hsa-miR-30a-3p, and hsa-miR-30e-3p, while restricting the analysis to genes associated with scoliosis and osteogenesis. The results, depicted in Figure 6, demonstrate the identified gene-miRNA interactions.

### 2.3. Isolation and Characterization of Circulating Extracellular Vesicles

Extracellular vesicles (EVs) were purified by plasma of severe females AIS patients (*n* = 5) and female HC (*n* = 5). EVs were characterized by Western blot analysis (Figure 7A), confirming the expression of specific exosomal markers, such as Alix and CD63. Dynamic light scattering analysis permitted to accurately assess their size distribution, showing an average hydrodynamic diameter of ~160 nm (Figure 7B).

### 2.4. Circulating Extracellular Vesicles from Severe Female AIS Inhibit Osteogenic Differentiation

We investigated the role of AIS-EVs from severe females in hMSC osteogenic differentiation, a cellular mechanism dysregulated in AIS. qRT-PCR analysis validated the expression of miR-30a-5, miR-30a-3p, miR-30d-5p, and miR-30e-3p in AIS-EVs (Figure 8A); in addition, the expression of miR-30 family members increased in hMSC after 24 h of treatment (Figure 8B). We also analyzed the expression of *SMAD1*, *SMAD2* and *SMAD5*, and *BECN1* (Appendix A); these latter genes were regulated by miR-30 family members and were described as genes involved in signaling pathways crucial for osteogenesis. As shown in Appendix A, the expression of *BECN1* decreased in hMSC treated with AIS-EVs for 14 days, both in the basal and differentiation medium, whereas *SMAD5* decreased only in the differentiation medium. We then evaluated the expression of osteogenic differentiation markers; as shown in Figure 8C,D, AIS-EVs treatment negatively affected the expression of *ALPL* and *RUNX2* in hMSC cultured in both basal and differentiation medium for 14 days.

Moreover, we demonstrated a relevant effect of AIS-EVs on the expression of both *COL1A1* RNA and secreted protein, which resulted in strongly down-regulated expression (Figure 9A,B). Alizarin red staining, commonly used to identify and stain calcium deposits, revealed a significant delay in osteoblastic mineralization when hMSCs were treated with AIS-EVs (Figure 9C).

### 2.5. Proteomic Analysis of AIS-Derived Extracellular Vesicles 

The proteome of AIS-EVs (*n* = 5 females with severe AIS) and HC-EVs (*n* = 5 females HC) was analyzed using mass spectrometry (MS)-based proteomics, following the methods described in materials and methods. It should be noted AIS-EVs exhibited a few dysregulated proteins, which were found to be associated with osteogenesis imperfecta, inhibition of osteogenesis and activation of adipogenesis. Figure 10A,B display the volcano plot and heat map, respectively, which report the significantly and differentially expressed proteins in both groups of analysis. Please refer to Appendix A for complete proteomic analysis.

## 3. Discussion

AIS represent the most common spinal deformity during puberty, occurring in young aged between 10 to 18 years old, characterized by growth spurts and changes due to puberty. It is defined by abnormal skeletal growth, impaired bone density and mineral metabolism [60]. Gender affects it, with a higher percentage in girls as they become older and with higher Cobb angle. Furthermore, genetic factors and age of onset are important determinants in AIS [8]. It is noteworthy adolescents with scoliosis experience both physiological and psychosocial consequences, including deformity, pain, and decreased ventilatory capacity as the thoracic cavity decreases. Consequently, AIS can have a negative impact on health-related quality of life and self-image [61,62]. It is therefore clear, an early analysis can positively influence future orthopedic choices, significantly reducing the possibility of surgery [63,64]. This could also be made possible by means of biomarkers with diagnostic and prognostic significance; the identification of these candidates could improve disease management and also provide insight into the pathogenesis of AIS.

Circulating miRNAs, whether free or embedded in EVs, are considered promising candidates for biomarkers discovery due to their finely tuned dysregulation during diseases progression, representing a hallmark also in orthopedic disorders [33,65,66,67]. Furthermore, minimally invasive extraction procedures clearly benefit the recovery of circulating miRNAs from blood [68,69]. Several studies have identified a significative correlation between changes in circulating miRNA levels and the risk of osteoporosis and fragility fractures [70,71,72]. A specific signature of circulating miRNAs seems to reflect the presence of osteoporotic vertebral fractures in postmenopausal women, while circulating miRNAs are correlated with the levels of bone turnover markers[72]. Even in intervertebral disc degeneration (IVDD) were identified specific circulating miRNAs, undergoing perturbation during the pathogenesis of disc degeneration [43]. Bioinformatic analyses have confirmed these circulating miRNAs, identified in orthopedic pathologies, were involved in multiple signaling pathways associated with pathogenesis. These miRNAs can functionally participate in the initiation and progression of orthopedic disease, can regulate osteogenic differentiation or bone formation and skeletal muscle [73].

Recent studies have revealed circulating miRNAs acted as a distinct biomarker signature to diagnose AIS with high sensitivity and specificity [22]. Researchers demonstrated an active role of circulating miRNAs in the epigenetic control of signaling pathways by regulating osteoblast and osteoclast differentiation. Moreover, some of them were already known for their involvement in cell signaling pathways that controlled the pluripotency of stem cells, such as the Wnt/β-catenin pathway [22]. Other studies compared data from differentially expressed miRNAs on bone tissues with the plasma levels of the selected miRNAs candidates, to develop a composite diagnostic model for AIS [50]. Overexpression of miR-151a-3p in primary osteoblasts from severe AIS patients negatively affected bone homeostasis, and the high plasma expression levels suggested miR-151a-3p as a potential biomarker for severe AIS. Furthermore, the authors did not find any dysregulated miRNAs that could be indicated as potential biomarkers for moderate AIS, hypothesizing dysregulated miRNAs were mainly involved in the progression of unbalanced spinal growth. [23]. Thus, despite the complexity of the disease, the role of miRNAs in the pathogenesis of AIS is emerging. 

In this study, to investigate these aspects as well, we performed miRNAs profiling from 20 plasma samples of AIS patients and 10 plasma samples from HC, using microfluidic cards. The results showed a group of miRNAs that were dysregulated in the AIS patients compared to HC. Bioinformatic analysis revealed the differentially expressed miRNAs targeted genes involved in cell signaling pathways that regulated bone homeostasis and also suggested an association with several diseases, including scoliosis, spinal curvature, and other bone characteristics. Furthermore, some miRNAs had already been described for their role in other orthopedic pathologies[51,52,74,75,76]. Three miRNAs were down-regulated in our AIS patients: miR-1294, miR-200a, and miR-548m. Interestingly, it is known miR-1294 targets HOXA9 and has a tumor suppressive role in osteosarcoma where it is down-regulated, correlating with a poor 5 year overall survival [77,78]. MiR-200a activates Nrf2 (NF-E2-related factor: a key antioxidant signaling) in primary human osteoblasts, protecting them from dexamethasone. Oxidative stress is a primary contributor to orthopedic ailments, including osteoarthritis and osteoporosis. The stress interferes with bone remodeling and increases the likelihood of fractures [79,80]. Severe muscle injury and accumulated oxidative stress increased in idiopathic scoliosis compared to HC [81]. Finally, in tumour microenvironments, the down-regulation of miR-548m has been linked with the activation of c-Myc, which subsequently up-regulated HDAC6, resulting in stroma-mediated survival of cancer cell. 

It should be noted inhibiting histone deacetylase promoted osteoblast maturation and the use of HDAC inhibitors have been recently proposed for the treatment of bone disease [82,83,84,85]. The results of our study aligned with previous research on these three microRNAs, leading us to hypothesize they could be considered a unique microRNA signature of AIS disease. 

However, our study revealed a significant portion of the dysregulated circulating miRNAs was overexpressed in the plasma of AIS patients compared to HC. Of all the miRNAs evidenced in AIS patients, a select few had particularly grasped our attention more thoroughly, such as a group of circulating miRNAs (hsa-miR-30a-3p, hsa-miR-30a-5p, hsa-miR-30e-3p, hsa-miR-30d) belonging to the miR-30 family. Numerous studies highlighted the close relationship between the miR-30 family and osteogenesis and may be implicated in the occurrence and development of bone and joint diseases [51]. Their role is described in several orthopedic diseases, such as osteoporosis [86,87], arthritis [88,89], intervertebral disc degeneration [90,91], and bone tumors [92]. Critical factors, such as Smad1 and Runx2, are common target genes of miR-30 family members, which actively regulate signaling pathways of bone homeostasis (Wnt/β-Catenin, mTOR, PI3K/AKT, etc.) [51]. Even EVs, circulating in blood and acting as vehicles between cells, can modify the phenotype of target cells by packaging and delivering miRNAs belonging to miR-30 family. For example, circulating EVs secreted by neoplastic mast cells deliver miR-30a in osteoblast and inhibit differentiation, by suppressing Runx2 and SMAD1/5 expression. They can induce mineralization decrease and drastically reduce trabecular bone volume and microarchitecture, in mice [55]. Notably, overexpression of miR-30b-5p and miR-30c-5p was observed in a small cohort of patients with severe scoliosis compared to healthy controls, confirming for us the potential importance of this family of miRNAs in severe AIS [23].The importance of miR-30 family in bone led us to carry out further analysis in specific groups of AIS patients (severe female AIS patients vs. female HC; moderate/mild AIS patients vs. HC; male AIS patients vs. male HC). Our results indicated for the first time circulating miR-30 members were significantly overexpressed only in female patients with severe scoliosis. These patients were characterized by high Cobb angles and many of them underwent surgery. Conversely, AIS patients with moderate or mild scoliosis, as well as severe male patients, did not present a significative up-regulation of miR-30 members, when compared to HC. Overall, the data collected indicated circulating miRNAs belonging to the miR-30 family could be promising candidates for discriminating severe AIS patients from HC. Future analyses are needed to examine the expression of miR-30 members in larger cohorts of patients, adequately representing all degrees of disease, from mild to severe.

MSCs play a significant role in the etiology and pathogenesis of AIS; several studies report a possible mechanism leading to low bone mass in AIS can be due to a reduced osteogenic differentiation ability of hMSCs and a decreased bone mineral density [25,53]. This information, together with the evidence that circulating EVs in pathological conditions could alter bone homeostasis [93], led us to investigate the role of circulating AIS-EVs in the osteogenic differentiation. In particular, we chose to treat MSCs with EVs from severe AIS females because of the results of the previous analyses. Indeed, we know EVs contain regulatory molecules, such as ncRNAs and proteins, which they deliver to target cells to influence their activity [94]. We confirmed the expression of miR-30a-3p, miR-30a-5p, miR-30e-3p, and miR-30d in AIS-EVs from severe females and then we demonstrated the expression of miR-30 family members increased in hMSCs treated with circulating AIS-EVs from severe females. Moreover, we observed inhibition of osteogenic differentiation when hMSCs were treated with AIS-EVs; in particular, we found a decreased expression in markers for osteogenic differentiation and a reduced secretion of Col1A1, the most abundant ECM protein highly implicated in disease [95]. The data obtained allowed us to hypothesize the molecules packaged in AIS-EVS could negatively influence the osteogenic differentiation of hMSC, partly also through the specific miRNA content. Conversely, EVs from healthy subjects did not affect osteogenesis. Finally, since EVs contained a large repertoire of regulatory proteins that, together with the ncRNAs component, transmitted specific signals to their target cells, we decided to also investigate the protein component of AIS-EVs from severe females and HC. The proteomic analysis of EVs from plasma samples of severe AIS females revealed a small group of dysregulated proteins. Specifically, we observed elevated levels of amyloid A protein (SAA1) and cofilin 1 (CFL1) in AIS patients compared to HC. SAA1 is a proinflammatory adipokine described in human osteoarthritic (OA) joints, resulting from a local production, and diffusion due to abnormally high plasma concentration. It has been demonstrated the A-SAA expression within the joint cavity of OA patients may play a key role in inflammatory process associated with osteoarthritis [96]. In addition, A-SAA levels in OA plasma and synovial fluid are positively correlated with the Kellgren and Lawrence grade, which measures the severity of the disease [96]. In mice, the high production of hepatic serum amyloid A1 promoted chronic inflammation and caused bone loss [97]. Regarding cofilin-1, a recent pilot study reported adipokines were implicated also in AIS development and progression [98]. Bisphosphonates disrupts actin organization and decreases in vitro cofilin levels, an actin regulatory protein that severs actin filaments [99]. A pharmacological decrease in cofilin caused an inhibition of prostate cancer cells invasion, reducing the risk of bone metastasis. Studies have also shown inhibiting the expression of actin depolymerization factors enhances osteogenesis [99,100,101]. Lastly, it is important to note distinctive cytoskeletal alterations are responsible for osteogenesis imperfecta outcomes [101]. Data provided by the literature, together with our results, are very encouraging as they allow us to hypothesize the existence of new protein markers related to AIS. Collectively, our results demonstrated circulating miRNAs in plasma of severe AIS, as well as circulating vesicular proteins, could potentially discriminate severe AIS patients from healthy controls and could be used as biomarkers, after appropriate future validation studies. Furthermore, our preliminary study suggests these dysregulated circulating miRNAs, free or embedded in EVs, may also be involved in the deregulation of bone homeostasis, which may also contribute to the pathogenesis of AIS.

## 4. Materials and Methods

### 4.1. Study Design and Population

This work is based on a non-profit observational clinical study, methodologically defined as a ‘non-pharmacological interventional study using human biological samples in vitro’. During the entire duration of the study (28 months), 20 AIS patients (in a ratio of 5:1 between females and males) and 10 healthy controls (in a ratio of 1:1 between females and males) were enrolled. The study protocol was designed and conducted according to the principles of ICH-GCP and the guidelines of the Declaration of Helsinki. The investigators were assisted by the Clinical Trials Center of the IRCCS Istituto Ortopedico Rizzoli (CTC.IOR) in the design of the study (protocols and methods), in the preparation of the documents required for submission to the Ethics Committee, in the administrative monitoring of the data, and in the management and archiving of the documents. The protocol approved by the CTC.IOR was submitted to the Ethics Committee ‘Comitato Etico di Aria Vasta Emilia-Romagna’ (Bologna, Italy), which evaluated the regulatory, legal, methodological, and ethical aspects to approve the study protocol (ID: CE-AVEC 575-2020-Sper-IOR). Patients and the control group (healthy volunteers) were enrolled in the clinical study solely after submitting written informed consent. Inclusion criteria for the patient group were: ages between 11–17 years; diagnosis of idiopathic scoliosis with a Cobb angle > 10°; minimum two years of follow-up of the pathology; clinical data and radiological tests available; and no surgical treatment prior to enrollment in the study. Inclusion criteria for healthy controls were: ages between 11–17 years; and healthy subjects not affected by orthopedic and oncological pathologies. Exclusion criteria for patients and healthy controls were: severe cognitive deficits or psychiatric disorders; patients with secondary scoliosis (for cases); and women pregnant. All patients and healthy controls were non-smokers. The inclusion and exclusion criteria were discussed and approved by the group of orthopedic surgeons, in agreement with the researchers, together with the CTC.IOR, before being positively evaluated by the ethics committee. The clinical study is a monocentric study. The patients were recruited and followed exclusively by the orthopedic surgery specialists of the Rizzoli Orthopedic Hospital, for the time required for the clinical study.

The sample size was calculated using Bioconductor’s ‘sizepower’ package (Sample Size and Power Calculation in Microarray Studies. v:1.40.0) considering a significance level α = 0.05 and a power 1 − β = 0.90. The following parameters were considered for the sample size calculation: (1) hypothetical number of differentially expressed miRNAs equal to 8 out of 768 evaluated; (2) hypothetical number of false positives equal to 10% of the estimated; (3) effect size in miRNA expression equal to |log2(FC)| ≥ 1.5; (4) hypothetical standard deviation of the difference in miRNA expression between treatment and control equal to 0.98. The analysis estimated a minimum number of *n* = 10 patients per group. However, due to the complexity and heterogeneity of AIS, such as the risk of disease exacerbation varying according to degrees of Cobb observed at diagnosis, gender, familiarity, etc., the number of AIS patients was doubled to ensure the power of the pilot study.

### 4.2. Physical and Radiological Examinations

The physical examination entailed measuring various parameters, including age, gender, body mass index (BMI) in kg/cm^2^, the Lenke classification of scoliosis, the size of the curve determined through the Cobb angle measurement, and the Risser classification (Table 1). For enrolled patients, a radiological study was performed, comprising standing anteroposterior and lateral X-ray views (details of the radiological examinations are available upon request). X-rays images were not taken for HC. The clinical parameters collected enabled the classification of scoliosis, determination of the Cobb angle degree, Risser value, and classification according to the Lenke method.

### 4.3. Sample Plasma Collection

Whole human peripheral blood, collected in sterile BD Vacutainer Venous tubes containing the anticoagulant ethylenediaminetetraacetic acid (EDTA), was centrifuged at 2000× *g* for 15 min using a refrigerated centrifuge, within 2 h after collection. Plasma fractions were subsequently collected, aliquoted, and stored at −80 °C.

### 4.4. RNA Extraction and Quantifcation

Cell-free total RNA (including miRNAs) was isolated from plasma through TRIzol LS isolation with mirVana column clean-up method [102]. In detail, 350 μL of plasma was centrifuged for 20 min at 10,000× *g* at 4 °C, then plasma and TRIzol were mixed in a ratio of 1:3 and incubated at room temperature for 10 min. Two-hundred microliters of chloroform was added, followed by vortexing for 1 min and incubation at room temperature for 5 min. Phases were separated by centrifugation at 14,000× *g* for 15 min and aqueous layer was removed and mixed with 1.5 volumes of ethanol. The mixture was applied to mirVana kit filter columns followed by wash and elution steps according to the manufacturer’s protocol (Thermo Fisher Scientific, Waltham, MA, USA). RNA eluted with RNAse-free water was quantified in NanoDrop ND 2000 UV-spectrophotometer (Thermo Scientifc, Wilmington, DE, USA). In short, 3 μL of RNA (10–350 ng) were reverse transcribed by using the TaqMan^®^ microRNA Reverse Transcription Kit (Thermo Fisher Scientific, MA, USA) and the Megaplex RT Primers. A total of 2.5 μL of the reverse transcription product were used for the pre-amplification using the TaqMan^®^ PreAmp Master Mix (Thermo Fisher Scientific, MA, USA) and the Megaplex PreAmp Primers. For qRT-PCR, the pre-amplified product was mixed with “TaqMan^®^ Fast Advanced Master Mix” (Thermo Fisher Scientific, MA, USA) and the appropriate amount of H_2_O, and subsequently, loaded into the ports of the TaqMan array card (TaqMan™ Array Human MicroRNA B Cards v3.0, Thermo Fisher Scientific, MA, USA) (2 loading ports to cover the total of 96 reaction chambers). RealTime PCR was performed on a 7900HT Fast Real-Time PCR System (with TaqMan^®^ Array Block) (Thermo Fisher Scientific, MA, USA), using universal cycling conditions (92 °C/10 min, then [97 °C/1 s, 60 °C/20 s] for 40 cycles). The expression level of each miRNA was determined with the equation 2^−ΔΔCT^, using U6 snRNA as housekeeping gene. 

### 4.5. Cell Culture

Human Mesenchymal Stem Cells (hMSCs) were obtained from Lonza (Lonza, Walkersville, MD, USA) and grown in Mesenchymal Stem Cell growth medium (MSCGM BulletKit™, Lonza, Walkersville, MD, USA) to maintain them into an undifferentiated condition and in Mesenchymal Stem Cell Osteogenic Differentiation Medium to induce osteogenic differentiation (MSC Osteogenic Differentiation BulletKit™, Lonza).

### 4.6. Extracellular Vesicles Isolation 

Extracellular vesicles (EVs) from frozen plasma of AIS patients (*n* = 5) and HC (*n* = 5) were extracted by ‘Total Exosome Isolation Kit (from plasma)’ (Invitrogen, Thermofisher, MA, USA), according to manufacturer’s instructions after thawing on ice and centrifuging at 13,000 rpm for 1 min to separate cellular debris. EVs protein content was determined by the Bradford assay.

### 4.7. Nanoparticle Tracking Analysis (NTA)

Concentrations and size distribution of AIS-EVs and HC-EVs were measured by Nanoparticle Tracking Analysis (NanoSight NS300, Malvern Instruments Ltd., Malvern, UK). Samples were diluted in phosphate buffered saline (PBS) 1:100 to reach optimal concentration for instrument linearity. The particle size measurement was calculated on a particle-by-particle basis in 3 videos of 60 s to provide accuracy and statistics for each analysis, under the following condition: cell temperature: 23.3°–23.6 °C; Syringe speed: 30 µL/s. Recorded data were analyzed for the mean, mode, median, and estimated concentration of particles by the in-build NanoSight Software NTA 3.3. Hardware: embedded laser: 45 mW at 488 nm; camera: sCMOS.

### 4.8. Human Mesenchymal Stem Cells Treatment with EVs

hMSCs were plated in 6-well plates in the basal Mesenchymal Stem Cell growth medium; 24 h after seeding, when cells were 70% confluent, hMSCs were cultured with Mesenchymal Stem Cell Osteogenic Differentiation Medium or, alternatively, with the basal Mesenchymal Stem Cell growth medium. Cells were treated for 14 days with EVs (25 μg/mL) from severe AIS females or from HC to evaluate the effect of EVs on the osteogenic differentiation of hMSCs, compared to the untreated hMSCs. Every 3 days, the culture media, including EVs, was changed. Cells were also treated for 24 h with EVs from plasma samples to further evaluate the miRNA expression levels.

### 4.9. RNA Extraction and Real-Time PCR

Total RNA from hMSCs was extracted using TRIzol Reagent, (Invitrogen, Life Technologies, Carlsbad, CA, USA) according to the manufacturer’s protocol. qRT-PCR was used to confirm the expression levels of mRNAs. For mRNA detection, oligo-dT-primed cDNA was obtained using the High-Capacity cDNA Reverse Transcription Kits (AB SCIEX, Foster City, CA, USA) and then used as template to quantify mRNA levels by Fast SYBRR Green Master Mix (AB Applied Biosystem, Beverly, MA, USA). Relative changes in gene expression between hMSCs treated with AIS-EVs and HC-EVS samples were determined with the ΔΔCt method. Final values were expressed as fold of induction. 

For miRNA detection, total RNA was purified from hMSCs treated with AIS-EVs and HC-EVs using TRIzol Reagent, (Invitrogen, Life Technologies, Carlsbad, CA, USA) according to the manufacturer’s protocol. Total RNA from AIS-EVs was purified using Total Exosome RNA and Protein Isolation Kit (Invitrogen, Life Technologies, Carlsbad, CA, USA), according to the manufacturer’s protocol and stored at −80 °C until further use. The quantity and purity of total RNA were checked by Qubit RNA broad range assay (Invitrogen, Life Technologies, Carlsbad, CA, USA) using Qubit 2.0 fluorimeter (Invitrogen, Life Technologies, Carlsbad, CA, USA) and Nanodrop 1000 spectrophotometer (Thermo Fisher Scientific). Total RNA was converted to cDNA using TaqMan Advanced miRNA cDNA Synthesis Kit (AB Applied Biosystem, Beverly, MA, USA), according to the manufacturer’s protocol. The resulting cDNA was used for quantifying the expression levels of miRNAs using TaqMan Advanced miRNA Assays (Catalogue no. A25576, Applied Biosystems) and TaqMan Fast Advanced Master Mix (Catalogue no. 4444557, Applied Biosystems). The relative expression of miRNA, from AIS-EVs, was calculated by the 2^−ΔCt^ method after normalizing with miR-16a-5p. Relative changes in miRNA expression between hMSCs treated with AIS-EVs and HC-EVS samples were determined with the ΔΔCt method. Final values were expressed as fold of induction. 

### 4.10. ELISA Assay

Human collagen, type I, alpha 1 (COL1A1) level secreted by hMSCs, treated or not with circulating EVs from AIS or HC, were quantified by ELISA Kit for COL1A1Sandwich ELISA (Cloud-Clone Corp., Houston, TX, USA). Briefly, hMSCs were treated with circulating EVs (25 μg/mL) from AIS or HC and cultured, in basal or differentiation medium. Supernatants were harvested after 14 days and analyzed, according to the manufacturer’s protocol.

### 4.11. Western Blotting

Total proteins (30 µg per lane) from AIS-EVS and HC-EVs were analyzed by SDS-PAGE followed by Western blotting. Cell lysates were separated using Bolt Bis-Tris gel 4–12% (Thermo Fisher Scientific, Cambridge, MA, USA) and transferred through iBlot 2 Western Blot Transfer System (Thermo Fisher Scientific, Cambridge, MA, USA); the membrane was incubated in blocking solution (5% BSA, 20 mM Tris, 140 mM NaCl, 0.1% Tween) and probed overnight with the specific antibodies. The antibodies against the following proteins were used: Alix (3A9) from Cell signaling (Beverly, MA), CD-63 (sc-5275), and HSP 70/HSC 70 (sc-24) from Santa Cruz Biotechnology (Santa Cruz, CA).

### 4.12. Alizarin Red S Staining

After a 14 day osteogenic induction, hMSCs were fixed in 4% paraformaldehyde for 15 min, washed twice with deionized water and incubated with 1% of Alizarin Red S solution (Sigma-Aldrich, Milano, Italy), for 30 min at room temperature. After washing twice with PBS, digital images were captured using a Nikon Eclipse Ti microscope.

### 4.13. Proteomic Analysis

Sample preparation: Supernatant containing 30 μg of lysate EVs were collected and applied to filter aided sample processing (FASP)1. Briefly, proteins were reduced by the addition of 1 M Dithiothreitol (DTT, Thermo Fisher Scientific, MA, USA) in 100 mM Tris/HCl, 8 M urea pH 8.5 for 30 min at 37 °C. Proteins were then alkylated in 50 mM iodoacetamide (IAA, Thermo Fisher Scientific, MA, USA) for 5 min at room temperature and washed twice in 100 mM Tris/HCl, 8 M urea pH 8.0 at 14,000× *g* for 30 min. Proteins were digested with 0.2 μg LysC (Promega, Madison, USA) in 25 mM Tris/HCl, 2 M urea pH 8.0 overnight and with 0.1 μg trypsin (Promega, Madison, USA) in 50 mM ammonium bicarbonate for 4 h. Resulting peptides were desalted by stop-and-go extraction (STAGE) on reverse phase C18 (Supelco Analytical Products, part of Sigma-Aldrich, Bellefonte, US), and eluted in 40 μL of 60% acetonitrile in 0.1% formic acid. Then, volume was reduced in a SpeedVac (Thermo Fisher Scientific, MA, USA) and the peptides resuspended in 20 μL of 0.1% formic acid. Peptide concentration was measured by a NanoDrop microvolume spectrophotometer (Thermo Fisher Scientific, MA, USA) and 1 μg peptides were applied to LC-MS/MS analysis. 

LC-MS/MS: To achieve high sensitivity, a nanoLC system (Vanquish Neo UHPLC–part of Thermo Scientific) using an Acclaim PEPMap C18 column (25 cm × 75 µm ID, Thermo Scientific, Waltham) was coupled online to an Exploris 480 mass spectrometer (Thermo Fischer Scientific). Peptides were separated using a 130 min binary gradient of water and acetonitrile containing 0.1% formic acid. Data-independent acquisition (DIA) was performed using a MS1 full scan (400 *m*/*z* to 1200 *m*/*z*) followed by 60 sequential DIA windows with an overlap of 1 *m*/*z* and window placement optimization option enabled. Full scans were acquired with 120,000 resolution, automatic gain control (AGC) of 3 × 10^6^, and maximum injection time of 50ms. Afterwards, 60 isolation windows were scanned with a resolution of 30,000, an AGC of 8 × 10^5^ and maximum injection time was set as auto to achieve the optimal cycle time. Collision induced dissociation fragmentation was induced with 30% of the normalized HCD collision energy. 

The proteomic data were analyzed by the software DIA-NN (version 1.8.1) by using a predicted library generated from in silico digested human Uniprot reference database involving cuts at K* and R*, two missed cleavages allowed, minimal peptide length set a 6, which consisted of 20,923 proteins, 31,382 protein groups, 6,894,911 precursors in 2,140,232 elution groups.

### 4.14. Statistical Analysis

Statistical analysis was conducted using R software v.4.3.1 [2]. Continuous data (qRT-PCR and ELISA data) were reported as Mean ± 95%CI at a significant *p*-value < 0.05. After having verified the normal distribution of data (Shapiro-Wilk test), they were analyzed by two-sided Student *t* test (qRT-PCR data of miRNA) or by one-way ANOVA followed by Dunnett test (qRT-PCR and ELISA data of osteogenic differentiation markers).

Hierarchical clustering of the miRNA fold change was performed using Euclidean distance algorithms and a heat map was generated. The miRNA data were further analyzed with a two-sided Student’s *t*-test. The obtained *p*-values were adjusted for false discovery rate due to multiple testing correction [103]. Volcano plot analysis of dysregulated miRNAs was derived from plasma of AIS patients and HC (*p* ≤ 0.05 and fold change ≥ 1.5). MiRNET was employed to predict the target genes of the significantly up-regulated miRNAs. The miRNET database (https://www.mirnet.ca/faces/home.xhtml (accessed on 26 October 2023)) results from the integration of 11 existing miRNA–target prediction programs (TarBase, miRTarBase, miRecords, miRanda, miR2Disease, HMDD, PhenomiR, SM2miR, PharmacomiR, EpimiR, and starBase) [104]. 

Regarding proteomic data, the false discovery rate for peptide and protein identification was set at 0.01%. Label-free quantification (LFQ) was used for proteins quantification. The LFQ values were log2 transformed and a two-sided Student’s *t*-test was used to evaluate proteins statistically significantly regulated between the AIS-EVS and HC-EVs (*n* = 5) [105,106,107,108].

## 5. Conclusions

Our study highlighted the expression of a specific set of circulating miRNAs in severe AIS compared to HC. It also showed circulating EVs, with their content, can transmit molecular signals that contribute to the dysregulation of osteogenic differentiation, related to the pathogenesis of AIS. The results obtained point towards future studies, with a larger cohort of age- and sex-matched patients, in which the expression of a specific signature of miRNAs and the role of circulating EVs would be studied in depth and in relation to the severity and the pathogenesis of AIS.

## Figures and Tables

**Figure 1 ijms-25-00570-f001:**
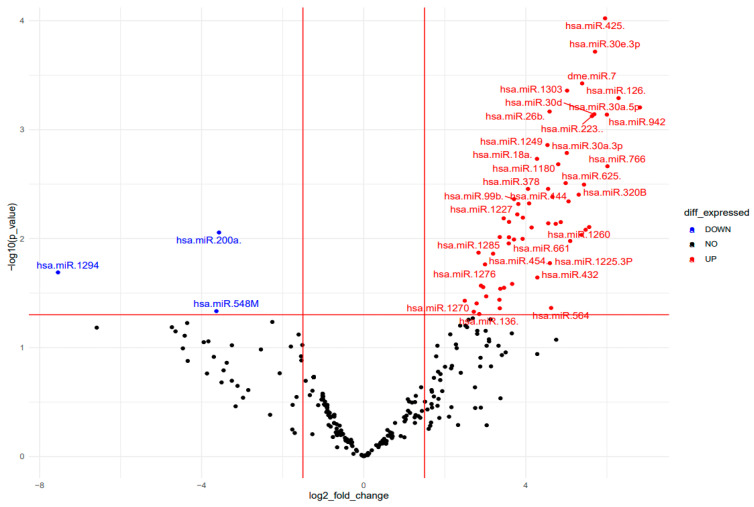
Differentially expressed miRNAs analysis in AIS patients and HC. Volcano Plot shows the relationship between fold change and *p*-values. The red and blue points in the plot represent the significantly up- and down- differentially expressed miRNAs, respectively.

**Figure 2 ijms-25-00570-f002:**
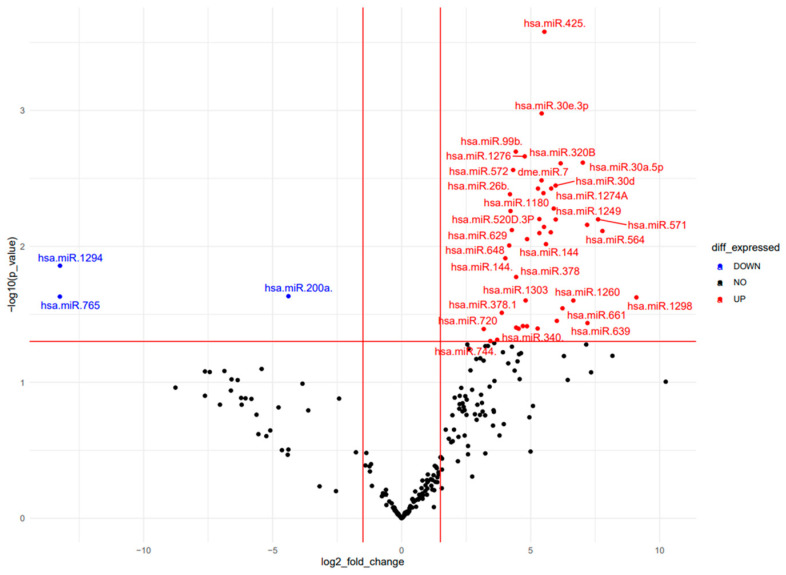
Differentially expressed miRNAs analysis in severe female AIS patients and female HC. Volcano Plot shows the relationship between fold change and statistical significance. The red and blue points in the plot represent the significantly up- and down- differentially expressed miRNAs, respectively.

**Figure 3 ijms-25-00570-f003:**
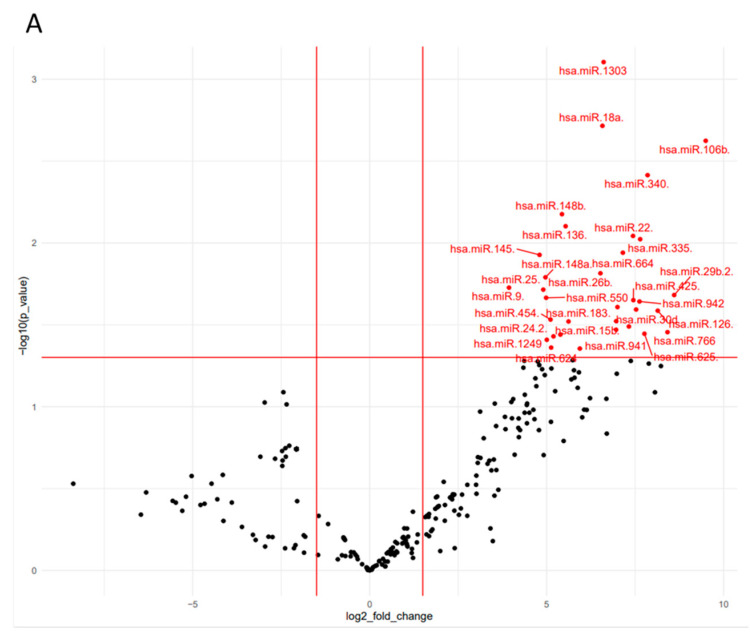
Differentially expressed miRNAs analysis in moderate AIS patients and severe male AIS patients. (**A**) Volcano Plot analysis for circulating miRNAs in moderate AIS patients vs. HC and (**B**) severe males AIS vs. males HC. The red and blue points in the plot represent the significantly up- and down- differentially expressed miRNAs, respectively.

**Figure 4 ijms-25-00570-f004:**
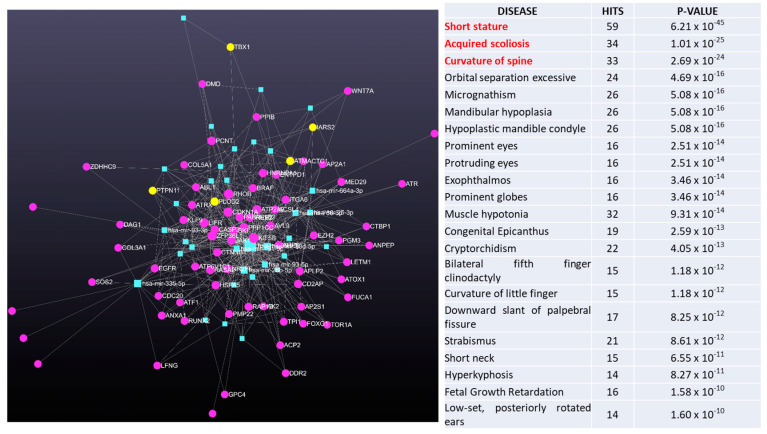
MiRNET was employed to predict the target genes of the significantly up-regulated miRNAs in AIS patients versus HC.

**Figure 5 ijms-25-00570-f005:**
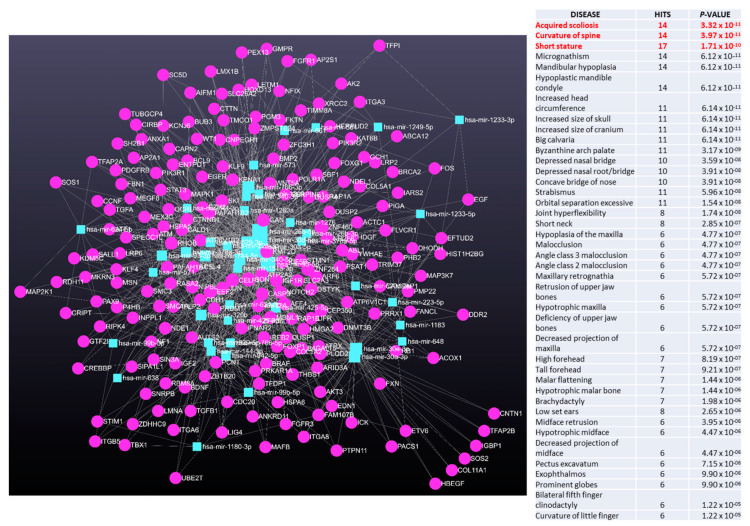
MiRNET was employed to predict the target genes of the significantly up-regulated miRNAs in severe females AIS patients versus females HC.

**Figure 6 ijms-25-00570-f006:**
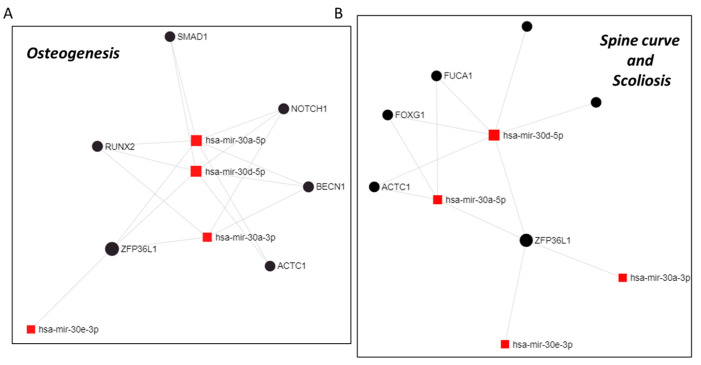
MiRNET analyses to predict the target genes of miR-30 family members (hsa-mir-30a-5p; hsa-mir-30d-5p; hsa-mir-30a-3p; and hsa-mir-30a-3p) in severe females AIS patients versus females HC. (**A**) miRNET analysis of miR-30-target genes in ‘osteogenesis’ and (**B**) miRNET analysis of miR-30-target genes in ‘Spine curve and Scoliosis’.

**Figure 7 ijms-25-00570-f007:**
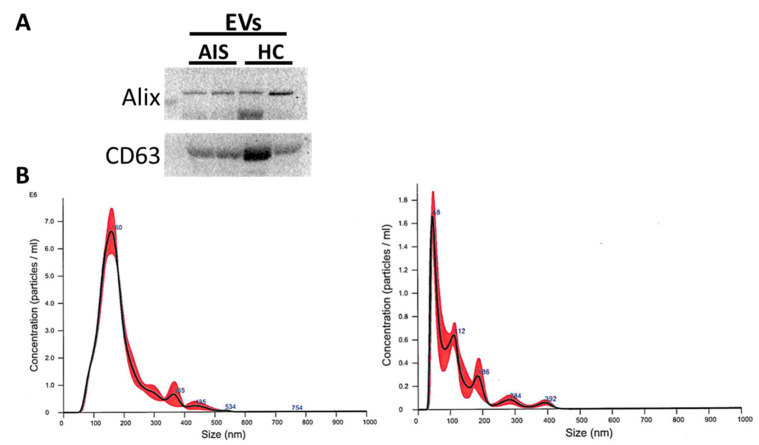
Characterization of circulating extracellular vesicles (EVs). (**A**) Western blot analysis of Alix and CD63 in circulating EVs lysates from 2 representative severe female AIS patients and 2 female HC. (**B**) Nanosight analysis of circulating EVs from 1 representative severe female AIS patient (left) and 1 female HC (right). Nanoparticle tracking analysis (NTA) confirmed the isolation of EVs from patients and HC, demonstrating the intrinsic heterogeneous characteristics of EVS.

**Figure 8 ijms-25-00570-f008:**
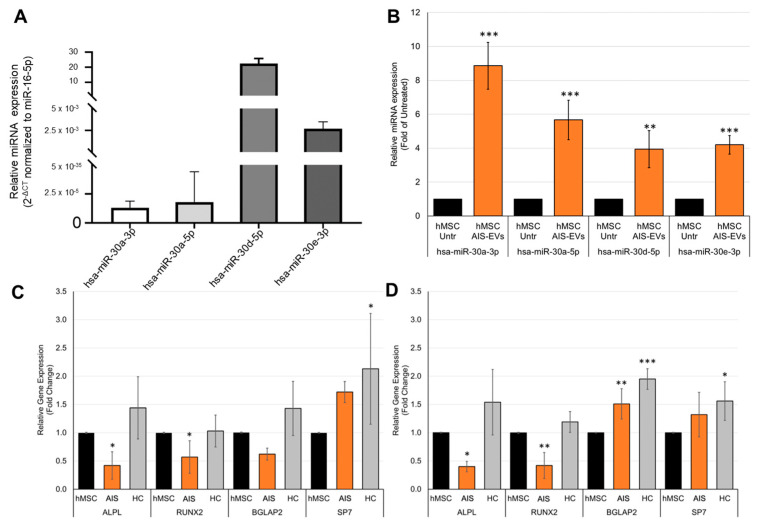
AIS-EVs reduce expression of osteogenic differentiation markers. (**A**) qRT-PCR results of the relative expression of miR-30d-5p, miR-30a-5p, miR-30e-3p, and miR-30a-3 in AIS-EVs expressed as 2^−ΔCt^ versus miR-16-5p. The Student *t* test showed a significant difference (*p* < 0.0005) for all miRNA values towards the average housekeeping expression value. (**B**) qRT–PCR analysis of miR-30d-5p, miR-30a-5p, miR-30e-3p, and miR-30a-3p in hMSCs untreated or treated for 24 h with AIS-EVs. Data were normalized for miR-16-5p and values were expressed as fold change in miRNA expression that occurred in hMSCs treated with AIS-EVs versus untreated hMSCs (Student *t* test: **, *p* < 0.005; ***, *p* < 0.0005). (**C**,**D**) qRT–PCR analysis of *ALPL*, *RUNX2*, *BGLAP2,* and *SP7* in hMSCs untreated or treated for 14 days with EVs from plasma of severe female AIS (*n* = 5) and plasma of female HC (*n* = 5), in basal (**C**) or differentiation (**D**) medium. Data were normalized for *GAPDH* and values were expressed as fold of control (untreated cells). One-way ANOVA followed by Dunnett test (treated groups vs. untreated group): *, *p* < 0.05; **, *p* < 0.005; ***, *p* < 0.0005.

**Figure 9 ijms-25-00570-f009:**
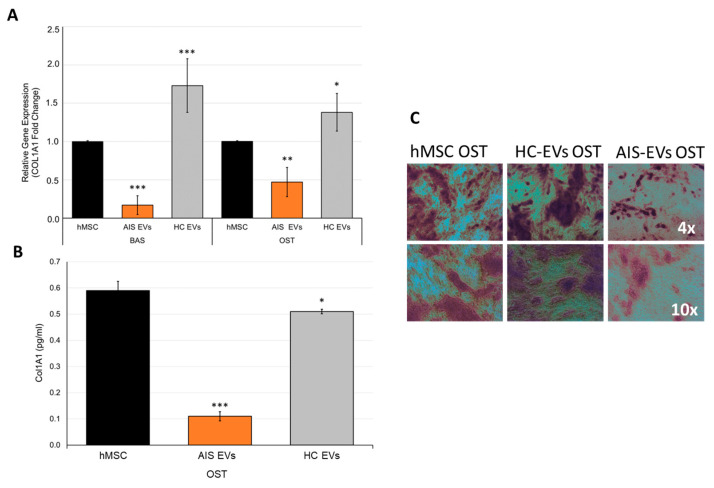
Circulating AIS-EVs from severe female patients reduce Col1A1 expression and reduce extracellular mineralization. (**A**) qRT–PCR analysis of *COL1A1* in hMSCs untreated or treated for 14 days with EVs from plasma of severe female AIS (*n* = 5) and plasma of female HC (*n* = 5), in basal (BAS) or differentiation (OST) medium. Data were normalized for *GAPDH* and values were expressed as fold of control (hMSCs BAS or hMSCs OST). One-way ANOVA followed by Dunnett test (treated groups vs. untreated group): *, *p* < 0.05; **, *p* < 0.005; ***, *p* < 0.0005. (**B**) Enzyme-linked immunosorbent assay (ELISA) of human COL1A1 protein levels in conditioned medium of hMSCs untreated or treated for 14 days with EVs from plasma of severe female AIS (*n* = 5) and plasma of female HC (*n* = 5), in differentiation medium. One-way ANOVA followed by Dunnett test (treated groups vs. untreated group): *, *p* < 0.05; ***, *p* < 0.0005. (**C**) Alizarin Red staining revealed a significant reduction in hMSC mineralization at day 14, after treatment with severe female AIS-EVs, compared to hMSC untreated or female HC-EVs treated.

**Figure 10 ijms-25-00570-f010:**
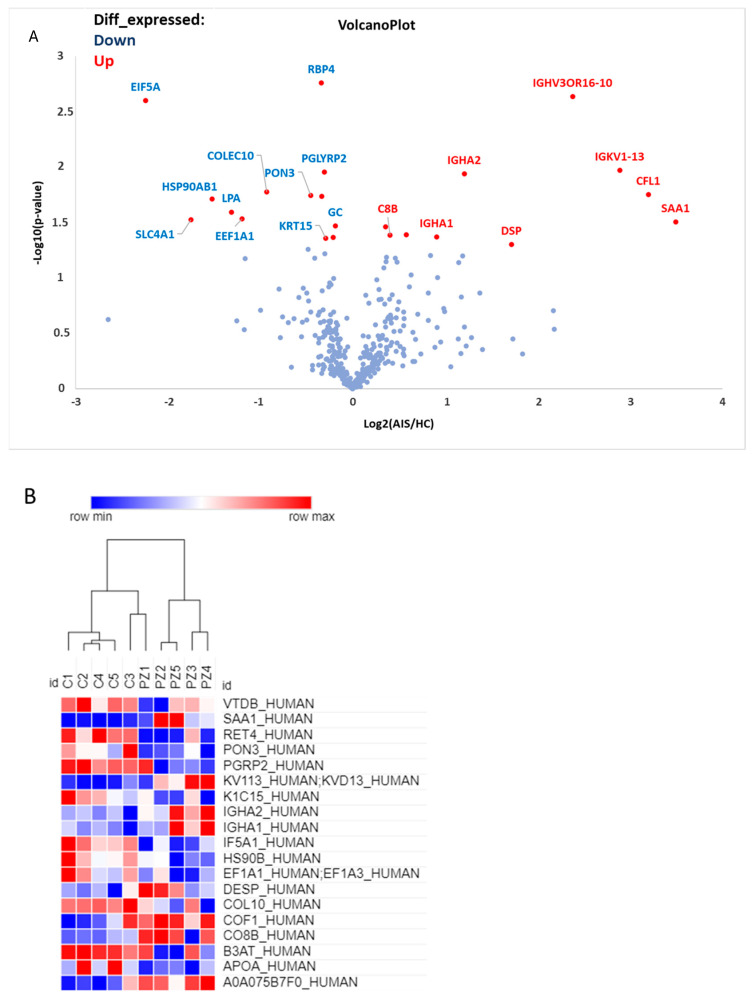
Proteomic analysis of AIS-derived extracellular vesicles. (**A**) Volcano plot shows the relationship between fold change and statistical significance. The names of the proteins are shown in red and blue in the plot. They represent respectively the significantly up- and down-regulated proteins. (**B**) Heat map diagram shows cluster analysis of differentially expressed proteins. The colors represent the expression values of differentially expressed miRNAs.

**Table 1 ijms-25-00570-t001:** Demographic and clinical parameters of adolescent idiopathic scoliosis patients; Mean value and 95% CI for age, Cobb and Risser angles, and BMI.

Variable	Mean [95% CI]	Range
AGE	14.7 [14.3, 15.0]	13–17
GENDER		
Male	3	
Female	17	
Cobb Angle (°)	54.6 [50.5, 58.7]	21–92
Risser Angle	3.4 [3.0, 3.7]	0–5
BMI	21.6 [20.6, 22.6]	16.4–33.2
LENKE CLASSIFICATION		
Type 1A	8	
Type 1B	1	
Type 1C	6	
Type 2A	1	
Type 3	1	
Type 5	2	
Type 6	1	

## Data Availability

The data presented in this study are available on request from the corresponding author.

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
