# Peer review of "Investigating the Differential Circulating microRNA Expression in Adolescent Females with Severe Idiopathic Scoliosis: A Proof-of-Concept Observational Clinical Study"

_ijms, 2024, doi:10.3390/ijms25010570_

Round 1
Reviewer 1 Report
Comments and Suggestions for Authors
The basis of this research is to determine specific biomarkers in plasma of AIS individuals and begin to elucidate a possible mechanism. The overall distribution of the data with 20 AIS individuals and only 10 HC and only three male patients who all had severe AIS made Initial data in figure 1 and figure 2 seem somewhat cherry picked. Later analysis better support the direction taken. Is it necessary to include male data analysis at all? What does it add to the story when it is underpowered and it isn't clear that significant differences could be found when examining so few individuals. My concern in including this data is that it may lead individuals to think there there are female specific changes where as there may just not be enough males in the study to see changes. Modify reporting or address in the text. Why was only miRnet software used? Specify in text what advantage this software had over other analysis tools or consider adding analysis with other tools to see how it compares. For example Target Scan. In figure 7 where the differences in the nano-sight analysis of circulating EVs meaningful? Were they consistent across multiple samples? Please address this in the text. How many samples were examined using Nano-sight analysis? In experiments where isolated EVs were added to hMSCs it is unclear if the EVs themselves contained the miR-30 family members or a signal that regulated family member expression. An easy way to address this would be to perform qPCR on the isolated EVs themselves. Overall this study should be of interest to individuals looking for biomarkers and possible mechanisms for further testing in Adolescent Idiopathic Scoliosis. Minor: The figure titles looking at differentially expressed miRNAs should indicate that the differential expression is based on microarray analysis. Because the methods are later this isn't abundantly clear looking at the figures.
Comments on the Quality of English LanguageAcceptable.
Author Response
Point by Point Response to Reviewer #1
The basis of this research is to determine specific biomarkers in plasma of AIS individuals and begin to elucidate a possible mechanism. The overall distribution of the data with 20 AIS individuals and only 10 HC and only three male patients who all had severe AIS made Initial data in figure 1 and figure 2 seem somewhat cherry picked. Later analysis better support the direction taken.
Is it necessary to include male data analysis at all? What does it add to the story when it is underpowered and it isn't clear that significant differences could be found when examining so few individuals.
My concern in including this data is that it may lead individuals to think there are female specific changes where as there may just not be enough males in the study to see changes. Modify reporting or address in the text.
We agree with the reviewer's comments that we should be cautious in stating that the expression of some miRNAs is exclusive to severe females. So far, we can say that we found deregulated miRNAs in 20 AIS individual compared to HC; notably, some miR-30 family members resulted overexpressed in severe females AIS compared to HC.
Nevertheless, the fact that the three males are severe patients and do not have overexpression of miR-30 family members is very intriguing. Interestingly, a sex difference in their expression has been reported, although in different diseases and/or human tissues (DOI:10.1101/2023.02.27.530361; PMID: 30382766; PMID: 38012706). The number of male participants was determined during the clinical protocol design, as described in the Methods section of the revised manuscript. Subsequently, our orthopaedic surgeons classified these male participants as severe patients after enrolment.
Therefore, in the revised manuscript, we reported that the main message is that the deregulation of microRNAs belonging to the miR-30 family is highlighted in severe AIS females. We also commented on the potentially interesting result obtained in the 3 severe males and stressed the importance of investigating this aspect in future studies.
Why was only miRnet software used? Specify in text what advantage this software had over other analysis tools or consider adding analysis with other tools to see how it compares. For example Target Scan.
We decided to use only miRnet because of its characteristics, which we describe below. These information has been incorporated into the text.
miRNet (www.mirnet.ca) is considered an easy-to-use web-based tool designed for creation, customization, visual exploration and functional interpretation of miRNA–target interaction networks (Fan, Y.; Xia, J. miRNet-Functional Analysis and Visual Exploration of miRNA-Target Interactions in a Network Context. Methods Mol Biol 2018, 1819, 215-233). By integrating multiple high-quality miRNA-target data sources and advanced statistical methods into a powerful network visualization system, miRNet allows researchers to easily navigate the complex landscape of miRNA–target interactions to obtain deep biological insights. Most importantly, the miRNET database (https://www.mirnet.ca/faces/home.xhtml) results from the integration of 11 existing miRNA–target prediction programs (TarBase, miRTarBase, miRecords, miRanda, miR2Disease, HMDD, PhenomiR, SM2miR, PharmacomiR, EpimiR, and starBase). Because it integrates more analysis tools, it is much more powerful.
In figure 7 where the differences in the nano-sight analysis of circulating EVs meaningful? Were they consistent across multiple samples? Please address this in the text. How many samples were examined using Nano-sight analysis?
Nanoparticle Tracking Analysis (NTA) analysis measures nanoscale particles in the range of 10nm-1000nm, using the properties of both light scattering and Brownian motion to obtain the size distribution of nanoparticles in liquid suspension samples.
Therefore, NTA provided us with information on the size distribution of plasma EVs from AIS and HC, isolated using a validated commercial kit for the isolation of plasma EVs.
In particular, we have used the generic term EVs to refer to isolated vesicles because they represent a heterogeneous population, unlike exosomes (a homogeneous population of vesicles), which must have specific purity and homogeneity characteristics according to the MISEV guidelines (The first Minimal Information for Studies of Extracellular Vesicles: doi.org/10.1080/20013078.2018.1535750).
For this reason, also considering the variability of biological samples, differences in size distribution can be observed. The aim of NTA analysis is not to compare samples, but to verify that the isolation procedure resulted in the isolation of EVs. The procedures for biological sample collection (from peripheral whole blood), storage, processing and analysis followed the guidelines of the MISEv guidelines, as discussed in previous studies (PMID: 31294446; PMID: 32759820).
We added this information in the caption of Figure 7.
In experiments where isolated EVs were added to hMSCs it is unclear if the EVs themselves contained the miR-30 family members or a signal that regulated family member expression. An easy way to address this would be to perform qPCR on the isolated EVs themselves.
In the first part of our study we demonstrated the deregulation of circulating microRNAs in severe females (among these, in particular, some of the miR-30 family members). Then, we focused our attention on plasma circulating EVs from severe AIS females. As it is known, circulating EVs are regulatory vesicles able to transmit their content (ncRNAs and proteins) to their target cells. Therefore, circulating EVs are considered potential biomarkers and they also have a role in the pathogenesis of disease.
Considering the deregulated expression profile of circulating miRNAs (free and vesicle-encapsulated) in females with severe AIS, we then investigated the potential role of the circulating EVs component in the pathogenesis of AIS; in particular, we examined the effects of AIS EVs from severe females on the osteogenic differentiation of MSCs. We hypothesised that the overexpression of miR-30 family members was not only due to the miRNA-free components, but also to the microRNAs embedded in EVs. In fact, several studies reported that EVs contain ncRNAs, can reflect the disease status, are stable and resistant structures that fully transmit their load to the target cells.
In Figure 8A of the old manuscript, we analyzed the expression of some miR-30 family members in hMSCs treated for 24 hours with AIS-EVS from severe females. The increased expression of miR-30 family members in hMSCs, compared to hMSCs untreated, leds us to hypothesize that these microRNAs have been transferred into the MSC.
Nonetheless, the reviewer's comment is correct; further analysis of these miRNAs in extracellular vesicles would confirm the presence of miR-30 family members.
For this reason, we performed a qRT-PCR analysis of miR-30a-5p, miR-30a-3p, miR-30d-5p and miR-30e-3p in AIS-EVs from severe females (Figure 8A of the revised manuscript)
New data and comments have been included in the revised version of the manuscript.
Regarding the qRT-PCR analysis of BECN1 and SMAD1-2-5, we have moved these results to SF4, and we have better commented on the significance of these analyses in the text of the revised manuscript.
Overall this study should be of interest to individuals looking for biomarkers and possible mechanisms for further testing in Adolescent Idiopathic Scoliosis.
Minor: The figure titles looking at differentially expressed miRNAs should indicate that the differential expression is based on microarray analysis. Because the methods are later this isn't abundantly clear looking at the figures.
We reported this information in the results section regarding this analysis.

Reviewer 2 Report
Comments and Suggestions for Authors
In this paper, the authors identified a first miRNA signature of global AIS, and then identified a restricted miRNA signature for severe AIS in female patients. They also explored the role of EVs in the osteogenic physiopathology. This paper is well written, and the results are clearly presented and very interesting. As a general comment, this paper shows many results that could have been presented in two different papers. First part of the results concern miRNA AIS signatures, and the second part relates to EV investigation. It looks like the authors missed to explain why they wanted to investigate EVs, and what the results of these investigations bring to their identified signature. Some elements explaining the scientific strategy should be added in the introduction and should be discussed with regard to the obtained results. Indeed, the objective of the EVs experiment is not clearly formulated in the discussion and is missing in the conclusion.
Major comments
1) Title. The title of the paper does not refer to the whole story. Only the severe AIS miRNA signature is mentioned, what about the global AIS signature and all the results about EVs?
2) General. Authors should use the term “specific” carefully, in the title and throughout the paper, considering that not all experiments have been made to establish the specificity of the signature for severe AIS vs other osteogenic pathologies for example.
Besides, an analysis of miRNAs deregulation between severe AIS and moderate/mild AIS should be presented in the results, to provide evidence of the miRNA signature ability to discriminate severe vs mild/moderate AIS & healthy controls.
3) General. A lot of results are shown in this paper, and it is not so easy to understand the relationship between the identified miRNA signatures and EVs and proteins results. The authors should clarify their rationale and why they wanted to investigate EVs effects on osteogenic differentiation in both introduction, abstract and discussion. Line 28 and line 91.
4) Discussion. Authors should not overinterpret their results. If the miRNAs of miR-30 family are “specific” of severe AIS, they can not be “promising candidate for discriminating AIS patients from HC” when they are unable to discriminate patients with moderate and mild AIS. Line 314.
5) Discussion. If authors want to present miR-30 family members as “indicators of disease severity”, they should present analysis showing the specificity of these miRNAs deregulation for severe AIS patients vs mild/moderate AIS patients. Line 316. (see also comment #2).
6) Mat & met 4.13 Statistical analysis. The choice of the housekeeping gene for miRNAs normalization is crucial for the robustness of the study. Authors should argue the use of U6 snRNA, especially when U6 snRNA is described as unstable in human plasma in the literature (Tang & al 2015, Different Normalization Strategies Might Cause Inconsistent Variation in Circulating microRNAs in Patients with Hepatocellular Carcinoma). Line 507.
7) Mat & met 4.8 RNA extraction and Real-Time PCR. Please precise the endogenous control. The manual seems to only recommend to “use a specific endogenous control” or “use global normalization”. Line 446.
8) Mat & met 4.4 RNA Extraction and quantification. Authors should provide the exact reference of the miRNA panel that was used in their study (TaqMan array card), to allow the reader to know which and how many miRNAs were investigated at first. Line 398.
9) Results 2.4 Circulating Extracellular Vesicles from Severe Animals AIS Inhibit Osteogenic Differentiation. The authors made the choice to analyze EVs from HC and severe AIS patients, but not at EVs from mild/moderate AIS patients. It would have been interesting to investigate also EV’s from mild/moderate AIS patients, regarding their effect on MSCs (miR-30 family expression and osteogenic differentiation potential). Line 181.
Minor comments
10) Figure 10A. Authors should highlight on the graph which proteins are upregulated and which ones are downregulated, as they have done for miRNAs expression volcano plot. (Figures 1-3). They also should discuss this information in the discussion.
11) Mat & met 4.8 RNA extraction and Real-Time PCR. As it appears in the result part, microRNAs were purified from hMSCs, not directly from EVs. Line 433.
12) Introduction. Some details should be added to help the reader to understand for which pathology miRNAs are proposed as biomarkers. Line 83.
13) Mat & met 4.1 Study Design & Population. I don’t understand how one of your exclusion criteria can be “women of childbearing age not pregnant” when most of your female patients are at childbearing age (>= 15 years old) and not pregnant. Line 360.
14) Some references should be added to convince the reader:
- Line 67-69
- Line 242-244
- Line 255-257
15) Some typography mistakes should be corrected:
- Line 88. Please suppress “(.”
- Line 242. Please suppress “(.”
- Line 282. Please suppress “(.”
- Line 285. Please correct “tha”.
- Line 345. Please correct “..”
- Line 278. Please correct “within2 h”.
- Line 382. Please correct “@l”.
16) Figure 9A. Please modify your graph legends to clarify that the three first bars refer to basal and the other ones to differentiation medium.
17) Supplementary figure 1A. A color legend is missing.
18) Supplementary figure 2A. A color legend is missing.
19) Supplementary table 4. Please correct the title of the column B.
Author Response
Point by Point response to Reviewer#2
In this paper, the authors identified a first miRNA signature of global AIS, and then identified a restricted miRNA signature for severe AIS in female patients. They also explored the role of EVs in the osteogenic physiopathology. This paper is well written, and the results are clearly presented and very interesting. As a general comment, this paper shows many results that could have been presented in two different papers. First part of the results concern miRNA AIS signatures, and the second part relates to EV investigation. It looks like the authors missed to explain why they wanted to investigate EVs, and what the results of these investigations bring to their identified signature. Some elements explaining the scientific strategy should be added in the introduction and should be discussed with regard to the obtained results. Indeed, the objective of the EVs experiment is not clearly formulated in the discussion and is missing in the conclusion.
We thank the reviewer for his comments. We added comments on the in vitro assay with plasmatic EVs from severe AIS in the ‘’Introduction’’ and ‘’Discussion’’ sections. We have also included the message from these experiments in the Conclusions section.
MAJOR COMMENTS
1) Title. The title of the paper does not refer to the whole story. Only the severe AIS miRNA signature is mentioned, what about the global AIS signature and all the results about EVs?
We chose a title which enclosed concisely the revelation of the study, as indicated by the guidelines of the IJMS for authors. For this reason we focused on the importance of the dysregulated expression of a group of microRNA signature characterizing patients with severe AIS, when compared to healthy controls.
2) General. Authors should use the term “specific” carefully, in the title and throughout the paper, considering that not all experiments have been made to establish the specificity of the signature for severe AIS vs other osteogenic pathologies for example.
In this study, the term ‘’specific’’ is used to distinguish the particular deregulation of some miRNAs (e.g. members of the miR-30 family) in the severe form of AIS, compared to HC. In fact it was not our intention to compare the expression of these miRNAs with other orthopedic pathologies.
We carefully read and eventually modified the text to verify whether some of our statements incorrectly declared the specificity of dysregulated miRNAs as exclusive to severe scoliosis vs. other orthopedic pathologies.
We have changed the title to avoid giving the wrong message, as suggested by the reviewer#2 and the reviewer#3:
INVESTIGATING THE DIFFERENTIAL CIRCULATING MICRORNA EXPRESSION IN ADOLESCENT FEMALES WITH SEVERE IDIOPATHIC SCOLIOSIS: A PROOF OF CONCEPT OBSERVATIONAL CLINICAL STUDY
Besides, an analysis of miRNAs deregulation between severe AIS and moderate/mild AIS should be presented in the results, to provide evidence of the miRNA signature ability to discriminate severe vs mild/moderate AIS & healthy controls.
The analyses we conducted are related to the clinical protocol, which only compares the results with those of healthy controls. The statistical calculations for significance and sample size were designed accordingly. Although the results of this pilot study are encouraging, further studies are needed to compare different groups of patients with varying degrees of disease severity.
3) General. A lot of results are shown in this paper, and it is not so easy to understand the relationship between the identified miRNA signatures and EVs and proteins results. The authors should clarify their rationale and why they wanted to investigate EVs effects on osteogenic differentiation in both introduction, abstract and discussion. Line 28 and line 91.
We have modified both the Introduction and the Discussion section, in the revised manuscript, to better explain the rationale of our study, the experimental approach and the studies known from the literature that led us to investigate these aspects. We have tried to better explain the significance of the experiments with circulating EVs, also in relation to the pathogenesis of AIS.
4) Discussion. Authors should not overinterpret their results. If the miRNAs of miR-30 family are “specific” of severe AIS, they can not be “promising candidate for discriminating AIS patients from HC” when they are unable to discriminate patients with moderate and mild AIS. Line 314.
We modified the sentence ''Overall, the data collected indicated that circulating miRNAs belonging to the miR-30 family are promising candidates for discriminating AIS patients from HC'' because it provides incorrect information.
In the REVISED MANUSCRIPT: ‘’Overall, the data collected indicated that circulating miRNAs belonging to the miR-30 family are promising candidates for discriminating AIS severe patients from HC’’
5) Discussion. If authors want to present miR-30 family members as “indicators of disease severity”, they should present analysis showing the specificity of these miRNAs deregulation for severe AIS patients vs mild/moderate AIS patients. Line 316. (see also comment #2).
We apologize for any confusion. At this time, we cannot confirm that miR-30 family members are disease indicators based on the data we have obtained. However, our analysis has shown that these miRNAs are upregulated in females with severe AIS compared to HCs.
Moving forward, we plan to conduct further analyses with new clinical protocols involving more patients with varying degrees of disease severity. The upcoming studies will determine if the deregulated miRNAs are significantly correlated with disease severity. This will be achieved by comparing various patient groups.
6) Mat & met 4.13 Statistical analysis. The choice of the housekeeping gene for miRNAs normalization is crucial for the robustness of the study. Authors should argue the use of U6 snRNA, especially when U6 snRNA is described as unstable in human plasma in the literature (Tang & al 2015, Different Normalization Strategies Might Cause Inconsistent Variation in Circulating microRNAs in Patients with Hepatocellular Carcinoma). Line 507.
We thank the reviewer for pointing out the normalization issue. It is very difficult to identify a suitable endogenous control miRNA from miRNA data and existing literature is full of experimental data showing that it does not exist a right endogenous control for all samples and all tissue. We are aware of the Tang & al 2015 paper, but it is important to note that they find U6 snRNA unsuitable as normalizer miRNA inpatients with Hepatocellular Carcinoma. A very interesting paper from Veryaskina et al 2022 (Reference Genes for qPCR-Based miRNA Expression Profiling in 14 Human Tissues) shows that the real goal is to find a reference gene tissue specific and even carcinoma specific when the studies are in cancer field. The reasons why we used U6 snRNA reside in the fact that this miRNA is still used as reference gene in bone field. Here are some references:
1) C. Beyer C. et al. Signature of circulating microRNAs in osteoarthritis Ann. Rheum. Dis., 74 (3) (2015), pp. 1-7, 10.1136/annrheumdis-2013-204698;
2) He X. et al. Identification and Characterization of MicroRNAs by High Through-Put Sequencing in Mesenchymal Stem Cells and Bone Tissue from Mice of Age-Related Osteoporosis. PLoS ONE, 8 (8) (2013), 10.1371/journal.pone.0071895;
3) Li- H. et al. MiRNA-10b reciprocally stimulates osteogenesis and inhibits adipogenesis partly through the TGF-β/Smad2 signaling pathway. Aging and Disease, 9 (6) (2018), pp. 1058-1073, 10.14336/AD.2018.0214;
4) Xu t. et al. Exosomal miRNA-128-3p from mesenchymal stem cells of aged rats regulates osteogenesis and bone fracture healing by targeting Smad5. J. Nanobiotechnol., 18 (1) (2020), pp. 1-18.
The other important reason why we used U6 snRNA is because the configuration of the microfluidic cards used it for the analysis; U6 snRNA was spotted in three different plate positions. This configuration was determined by the microfluidic selling company, not the researcher. Another internal control will be added for the analysis of a larger cohort, as this study is a pilot study.
7) Mat & met 4.8 RNA extraction and Real-Time PCR. Please precise the endogenous control. The manual seems to only recommend to “use a specific endogenous control” or “use global normalization”. Line 446.
We added the information.
8) Mat & met 4.4 RNA Extraction and quantification. Authors should provide the exact reference of the miRNA panel that was used in their study (TaqMan array card), to allow the reader to know which and how many miRNAs were investigated at first. Line 398.
We added the information in the ‘Materials and Methods’ section: TaqMan™ Array Human MicroRNA B Cards v3.0.
9) Results 2.4 Circulating Extracellular Vesicles from Severe Animals AIS Inhibit Osteogenic Differentiation. The authors made the choice to analyze EVs from HC and severe AIS patients, but not at EVs from mild/moderate AIS patients. It would have been interesting to investigate also EV’s from mild/moderate AIS patients, regarding their effect on MSCs (miR-30 family expression and osteogenic differentiation potential). Line 181.
We agree with the reviewer that the analysis of EVs from patients with mild disease could be interesting and be part of a future study including a larger cohort of patients.
In this study, we decided to focus on EVs from severe AIS females and HCs, also driven by the results obtained in the first part, which highlighted a peculiar expression of some circulating miRNAs in severe AIS females, compared to HC.
Data from the literature indicated that these miRNAs are crucial factors in bone metabolism, also in bone diseases. On the other hand, we know that circulating EVs contain miRNAs and are also involved in the pathogenesis of orthopaedic diseases. Therefore, we wanted to investigate the effect of EVs from severe AIS females on hMSCs differentiation. Interestingly, we found an inhibitory effects of AIS-EVs on hMSC osteogenic differentiation.
We analysed the expression of miR-30 family members in EVs from severe AIS females (revised manuscript, as requested by a reviewer) and also observed an increased expression of these miRNAs in hMSC after AIS-EVs treatment. We also found that compared to HC EVs, AIS EVs from severe females contained a small set of proteins that have been described in ortopedic conditions.
MINOR COMMENTS
10) Figure 10A. Authors should highlight on the graph which proteins are upregulated and which ones are downregulated, as they have done for miRNAs expression volcano plot. (Figures 1-3). They also should discuss this information in the discussion.
We modified the graph by indicating which proteins were upregulated and which were downregulated. In the discussion section, we discussed the two upregulated proteins that were more interesting for our study, also through information from the literature.
Future protein validation of these two proteins will be evaluated in a new clinical protocol.11) Mat & met 4.8 RNA extraction and Real-Time PCR. As it appears in the result part, microRNAs were purified from hMSCs, not directly from EVs. Line 433.
We corrected the sentence. We also added information about ‘’Total RNA from AIS-EVs’’ as we performed qRT-PCR analysis of miR-30 family members in AIS-EVs, as requested by a reviewer.
12) Introduction. Some details should be added to help the reader to understand for which pathology miRNAs are proposed as biomarkers. Line 83.
We added the information required in the Introduction section.
13) Mat & met 4.1 Study Design & Population. I don’t understand how one of your exclusion criteria can be “women of childbearing age not pregnant” when most of your female patients are at childbearing age (>= 15 years old) and not pregnant. Line 360.
The sentence has been changed as it is confusing to read. The information on the characteristics of the clinical protocol (Materials and methods section) has been discussed in more detail, as requested by another reviewer. In particular, we have reported the correct definition, exactly as described in the approved clinical protocol.
REVISED MANUSCRIPT: Women currently pregnant is an exclusion criterion.
14) Some references should be added to convince the reader:
We added some references.
- Line 67-69
- Line 242-244 (We also better explained the meaning of the phrase)
- Line 255-257
15) Some typography mistakes should be corrected:
- Line 88. Please suppress “(.”
- Line 242. Please suppress “(.”
- Line 282. Please suppress “(.”
- Line 285. Please correct “tha”.
- Line 345. Please correct “..”
- Line 278. Please correct “within2 h”.
- Line 382. Please correct “@l”.
We corrected typography mistakes.
16) Figure 9A. Please modify your graph legends to clarify that the three first bars refer to basal and the other ones to differentiation medium.
We corrected the graph legends.
17) Supplementary figure 1A. A color legend is missing.
We added the information.
18) Supplementary figure 2A. A color legend is missing.
We added the information.
19) Supplementary table 4. Please correct the title of the column B.
We corrected the title of the column B.

Reviewer 3 Report
Comments and Suggestions for Authors
Comments on the Quality of English LanguageModerate changes necessary.
Author Response
Point by Point Response to Reviewer #3
Firstly, I would like to congratulate the authors on their relevant and interesting work.
However, some points need to be addressed: It is advisable that the authors work with English editing services to improve the readability of the text.
Typos must also be corrected.
We thank the reviewer for her/his comments. Below we provide the requested information. We included the information in the Materials and Methods section (‘Study design and population’ paragraph) of the revised manuscript.
Title: It is advisable that the authors summarize the title and include the study design type in it.
We modified the title of the manuscript, thus including the study design type in it:
‘’Investigating the differential circulating microRNA expression in adolescent females with severe idiopathic scoliosis: a proof of concept observational clinical study’’
Introduction
We suggest that the authors summarize the introduction but at the same time, include more relevant information about similar genetic studies. We found several studies published in the literature in recent years regarding scoliosis and microRNAs. In this way, the authors should state clearly what is their research question in the last paragraph of the introduction, and describe what is new and original and new about their study that adds to the current knowledge.
We included, in the introduction section, similar studies regarding scoliosis and miRNA; we also better explained the study rationale and the importance of the results obtained, underlining nevertheless that it is always a preliminary study.
Methods
1) The timeline of the study, as well as the study design type should be specified.
The study is a no profit observational clinical study, methodologically defined as a ‘non-pharmacological interventional study using human biological samples in vitro’. The duration of the study was 24 months, extended to 28 months for reasons related to an initial slowdown in the recruitment of patients and healthy controls due to problems related to the COVID pandemic.
2) Was this study protocol previously validated?
The study protocol was designed and conducted according to the principles of ICH-GCP and the guidelines of the Declaration of Helsinki. The investigators were assisted by the Clinical Trials Centre of the IRCCS Istituto Ortopedico Rizzoli (CTC.IOR) in the design of the study (protocols and methods), in the preparation of the documents required for submission to the Ethics Committee, in the administrative monitoring of the data, and in the management and archiving of the documents. The protocol approved by the CTC.IOR was finally submitted to the Ethics Committee 'Comitato Etico di Aria Vasta Emilia-Romagna' (Bologna, Italy), which evaluated the regulatory, legal, methodological and ethical aspects in order to approve the study protocol (ID: CE-AVEC 575-2020-Sper-IOR).
3) For how long were patients recruited?
Patients, as well as healthy controls, were recruited within the 28 months of the duration of the study.
4) Please further specify inclusion and exclusion criteria and explain how were these criteria chosen.
Inclusion criteria for patients: Ages between 11-17 years; Diagnosis of idiopathic scoliosis with a Cobb angle > 10°; Minimum two years of follow-up of the pathology; Clinical data and radiological tests available and no surgical treatment prior to enrollment in the study;
Inclusion criteria for Healthy Controls: Ages between 11-17 years; Healthy subjects not affected by orthopedic and oncological pathologies. Exclusion criteria for patients and healthy controls: Severe cognitive deficits or psychiatric disorders; patients with secondary scoliosis (for cases); women pregnant. The inclusion and exclusion criteria were discussed and approved by the group of orthopedic surgeons, in agreement with the researchers, together with the CTC.IOR, before being positively evaluated by the ethics committee.
5) Please further describe the setting in which patients were recruited and how they were diagnosed, for how long they were followed up, by which types of specialists they were seen.
The clinical study is a monocentric study. The patients were recruited and followed exclusively by the orthopedic surgery specialists of the Rizzoli Orthopedic Hospital, for the time required for the clinical study.
6) L364: The authors mention “clinical trial”. Please further clarify the study design.
We thank the reviewer for this observation. The study was definied as: observational clinical study. The experimental design of the biological study involved collecting peripheral blood samples from two groups: patients with AIS (cases) and healthy subjects without orthopedic or oncological pathologies (control group). Plasma was isolated from the samples, and microRNAs were subsequently extracted and analyzed. The study aimed to identify new potential markers involved in the onset and progression of scoliosis among the microRNAs analyzed. Both the AIS patients and the control group were only enrolled in the clinical study after signing and dating the written informed consent form, together with the patient's legal representative.
7) Please provide a flowchart showing how many patients were included and excluded from the study.
As described before, in the timeline of the study, only patients with the criteria established and approved by the ethics committee were enrolled.
9) Was the Scoliosis Research Society (SRS) scoliometer used in this study?
During patient recruitment, the diagnosis of scoliosis was based on the method of measuring the Cobb angle on the radiogram, according to the SRS criteria.
10) For each genetic and laboratory analysis, the authors must explain the rationale behind, their initial hypothesis and the reason for running those specific tests.
We explained in the Introduction section the rational behind our initial hypotheis; the experimental approach chosen was described in the results section.
11) Please provide a full neurological examination description from all the patients. Please provide a more complete baseline characteristics table, including smoking status, neurological comorbidities, surgical history, infection history, neurological exam findings, secondary causes of scoliosis, family history of orthopedic conditions.
The patients and healthy controls had no neurological problems, cognitive deficits, psychiatric disorders, secondary causes of scoliosis, family history of orthopaedic conditions and histories of infections during the enrolment medical examination conducted by the orthopaedists. All patients and healthy controls were non-smokers. All this summary information has now been reported in the text of the results.
12) Please provide more information about the healthy controls as well as their baseline characteristics as mentioned above.
We have already reported the information from healthy controls above.
13) Was data normally distributed?
Before performing inferential analysis for continuous data (qRT-PCR and ELISA data) the Shapiro Wilk test was applied.
14) Was the power of the study calculated?
The sample size was calculated using Bioconductor's 'sizepower' package (Sample Size and Power Calculation in Microrarray Studies. v:1.40.0) considering a significance level α=0.05 and a power 1-β=0.90. The following parameters were considered for the sample size calculation: 1) hypothetical number of differentially expressed miRNAs equal to 8 out of 768 evaluated; 2) hypothetical number of false positives equal to 10% of the estimated; 3) effect size in miRNA expression equal to |log2(FC)| ≥ 1.5; 4) hypothetical standard deviation of the difference in miRNA expression between treatment and control equal to 0.98. The analysis estimated a minimum number of n=10 patients per group. However, due to the complexity and heterogeneity of AIS, such as the risk of disease exacerbation varying according to degrees of Cobb observed at diagnosis, gender, familiarity, etc., the number of AIS patients was doubled to ensure the power of the pilot study (Julien C et al. 2013).
15) We suggest that the authors work with a professional statistician to improve the reporting of data. For example, there are analyzes in the results that were not described in the methods and confidence intervals, not only p values should be reported. Graphs are out of scale.
We thank the reviewer for this important comment. We have carried out a new statistical analysis, which we have included in the Methods section. We have revised the text and captions to provide the necessary information.
16) Was ANOVA performed for the analysis in Figure 8?
We apologise for the mistake, but the data in Figures 8C and 8D were not analysed with ANOVA. We applied a one-way ANOVA followed by Dunnett's test for comparison with the control group. The statistics paragraph was implemented.
17) L159-160: Please provide the correlation analysis description and parameters.
We apologize for using the term 'correlate' incorrectly at lines 159-160. We did not perform a correlation analysis, but the network analysis of circulating miRNA data followed by a functional enrichment analysis using a hypergeometric algorithm with various database (e.g. KEEG) indicated an association with several diseases, including scoliosis, spinal curvature, and other bone characteristics. We have modified the sentence accordingly.
Results and discussion
The authors should improve reporting of their results from a statistical standpoint as mentioned above. Additionally, considering the abundance of genetic studies with miRNAs and scoliosis in the past years, it is advisable that the authors include a table in their study comparing their findings with those from the existing literature. In the discussion, the authors should emphasize what is relevant and new about their study.
We thank the reviewer for the suggestion. We have tried to make the required changes.
We have included new studies describing the role of microRNAs in scoliosis, and then highlighted the relevance of our study in a discursive manner.
We also reported in the Introduction section that the interest of our study was to investigate circulating miRNAs: 1) because of their main feature of being considered as potential biomarkers, as circulating miRNAS reflect the disease state, but also 2) because circulating miRNAs, free or inside EVs, can have a regulatory role in target cells and may provide inspiration for more in-depth studies.
Some of the new studies included in the revised manuscript, as requested by the reviewer, describe miRNA analyses in non-idiopathic scoliosis (e.g., neuromuscular, syndromic, congenital scoliosis) where the deregulation of miRNAs may depend on multiple factors to be considered and 2) the miRNAs examined are not circulating.
A comparison of studies on microRNAs (circulating or non-circulating) in scoliosis (idiopathic and non-idiopathic, adolescent and adult) will be the subject of a future systematic review, which may include our data.

Round 2
Reviewer 2 Report
Comments and Suggestions for Authors
The authors provided satisfactory answers to most of the reviewer’s comments. The manuscript is much clearer, especially with regard to the objective and the main conclusions of the study.
I still have a few questions and comments that should be addressed before final acceptance for publication. Please be aware that I could only access to author’s answers to my own comments, not other reviewer’s.
1) miRNA expression normalization. Authors used TLDA cards to perform their miRNA analysis on plasma samples form patients and used U6 snRNA for normalization. Although one would have used global normalization (Ct mean) based on the hundreds of detected miRNAs in their samples, U6 may be used as well, provided that its signal was reproducibly detected in the appropriate range of detection for plasma samples.
When looking at miRNAs in EVs, authors normalized their results with miR-16-5p as a reference gene, which has been used in numerous EV studies as well. However, why did the authors use miR-16-5p to normalize their data from MSCs following treatment with EVs, instead of U6, since, according to the authors’ reply, “U6 is still used as reference gene in bone field”?
2) in vitro EV assay. The results described in the in vitro EV assays are inconsistent.
- First, authors should add more details for the experimental conditions of their in vitro EV assay. In particular, what was the EV concentration used? For the 14-day differentiation assay, how long was the incubation time with EVs? Was medium refilled with fresh EVs during this time or at time of medium change?
- Also, since the authors purified EVs from severe AIS patients and HC samples for their in vitro work on MSCs, why did they look at the expression of miR-30 family members only in EVs from AIS samples? This would clearly provide insight as to whether miR-30 members are differentially expressed in AIS EVs compared to HC EVs.
- Similarly, it is essential to measure expression levels of miR-30 members in MSCs following incubation with HC EVs in order to ascertain any effect from AIS EVs.
- More generally, Fig 8 and suppl Fig 4 present results obtained from MSCs treated either only with AIS EVs (Fig 8B & Suppl Fig 4) or with AIS AND HC EVs (Fig 8C-D). It is therefore hard to conclude on the global effect of AIS EVs when they are not systematically compared with control HC EVs.
3) General. Authors added a serious number of references to their introduction and discussion parts. In particular, they highlight the studies by Wang et al (2020) and Chen et al (2022) who suggested circulating miR-151a-3p and miR-96-5p as potential biomarkers for severe AIS, respectively. How do the results of the present study compare with these publications? Do these 2 miRNAs (at least) behave similarly in terms of differential expression? Conversely, was the miR-30 family among previously published studies?
4) Graphs in Fig 8 and 9. Why are the results of Fig 8B different in the revised manuscript compared to original submission (e.g., level of miR-30a-3p expression histogram)?
Also, why are statistical results different between the 2 versions? If authors changed their methods of statistical analysis, please explain why. Homogenize the title size of graph axis in Fig 8A compared the other panels.
5) Figure 7. The title and values are missing/incomplete on the Y-axis of the right panel in Fig 7B.
6) TLDA analysis. Authors used the term “microarray” when referring to the TLDA analysis, though this is a microfluidic qPCR platform. Please modify accordingly (lines 128 & 509).
Also, the correct reference for TLDA cards is TaqMan™ Array Human MicroRNA A & B Cards v3.0 (line 748).
Author Response
RESPONSE POINT by POINT to REVIEWER
The authors provided satisfactory answers to most of the reviewer’s comments. The manuscript is much clearer, especially with regard to the objective and the main conclusions of the study.
I still have a few questions and comments that should be addressed before final acceptance for publication. Please be aware that I could only access to author’s answers to my own comments, not other reviewer’s.
1) miRNA expression normalization. Authors used TLDA cards to perform their miRNA analysis on plasma samples from patients and used U6 snRNA for normalization. Although one would have used global normalization (Ct mean) based on the hundreds of detected miRNAs in their samples, U6 may be used as well, provided that its signal was reproducibly detected in the appropriate range of detection for plasma samples.
We agree with the reviewer that the main point in using a housekeeping gene is the reproducibility and the signal levels across the samples. In our samples U6 snRNA was detected at CT average = 20.78 ± 2.34 that is an optimum range for plasma samples.
When looking at miRNAs in EVs, authors normalized their results with miR-16-5p as a reference gene, which has been used in numerous EV studies as well. However, why did the authors use miR-16-5p to normalize their data from MSCs following treatment with EVs, instead of U6, since, according to the authors’ reply, “U6 is still used as reference gene in bone field”?
In the first part of the study, we analysed the expression of a panel of miRNAs using microfluidics cards which used U6 snRNA for normalization. We also discussed that this miRNA is still used as reference gene in bone field. Then, we analysed and validated the expression of some miRNAs belonging to the miR-30 family by a single miRNA assay.
Specifically, as reported in the Materials and Methods section, we performed microRNA analysis using TaqMan Advanced miRNA Assays, which requires the TaqMan Advanced miRNA cDNA Synthesis Kit and is generally characterized by a different chemistry that allows the detection of multiple miRNAs from a single sample. The TaqMan® Advanced miRNA Assays User Manual states that the TaqMan® Advanced miRNA Assays do not detect snRNAs or snoRNAs. Therefore, it is recommended to use other endogenous controls for these assays, e.g. hsa-miR-16-5p and for this reason U6 was not suitable for the validation experiments because of the chemistry used.
2) in vitro EV assay. The results described in the in vitro EV assays are inconsistent.
- First, authors should add more details for the experimental conditions of their in vitro EV assay. In particular, what was the EV concentration used? For the 14-day differentiation assay, how long was the incubation time with EVs? Was medium refilled with fresh EVs during this time or at time of medium change?
We thank the reviewer for this comment; we have included this information as a new paragraph in the Materials and Methods section of the revised manuscript.
hMSCs Treatment with EVs: hMSCs were plated in 6-well plates in the basal Mesenchymal Stem Cell growth medium; 24 h after seeding, when cells were 70% confluent, hMSCs were cultured with Mesenchymal Stem Cell Osteogenic Differentiation Medium or, alternatively, with the basal Mesenchymal Stem Cell growth medium. Cells were treated for 14 days with EVs (25 μg/mL) from severe AIS females or from HC to evaluate the effect of EVs on the osteogenic differentiation of hMSCs, compared to the untreated hMSCs. Every 3 days, the culture media, including EVs, was changed. Cells were also treated for 24h with EVs from plasma samples to further evaluate the miRNA expression levels.
- Also, since the authors purified EVs from severe AIS patients and HC samples for their in vitro work on MSCs, why did they look at the expression of miR-30 family members only in EVs from AIS samples? This would clearly provide insight as to whether miR-30 members are differentially expressed in AIS EVs compared to HC EVs.
- Similarly, it is essential to measure expression levels of miR-30 members in MSCs following incubation with HC EVs in order to ascertain any effect from AIS EVs.
- More generally, Fig 8 and suppl Fig 4 present results obtained from MSCs treated either only with AIS EVs (Fig 8B & Suppl Fig 4) or with AIS AND HC EVs (Fig 8C-D). It is therefore hard to conclude on the global effect of AIS EVs when they are not systematically compared with control HC EVs.
We thank the reviewer for these observations/ questions; hereby we have provided our answers:
Figures 8A and 8B are a single qRT-PCR validation of the expression of circulating miR-30 (miR-30a-3p/5p; miR-30d-5p and miR-30e-3p) that were found to be overexpressed in females with severe AIS compared to healthy female controls (analysis using microfluidic cards, also see Suppl.Table 3). MicroRNAs expression was therefore not validated through single qRT-PCR in EVs from HCs because they were not significantly detected in HC samples.
Therefore, our aims were:
- to validate the expression of miR-30 in EVs from females with severe AIS.
This allowed us to hypothesize that packaged miRNAs can be transferred into their target cells (hMSCs) and partially explain their effect on processes such as osteogenesis (investigated in Figure 8C-D and Figure 9). We used the term 'partially' because the effects of AIS-EVs on osteogenesis differentiation are likely due to the specific content of the AIS-EVs. This content includes a combination of vesicular ncRNAs, with miR-30 being highlighted as over-expressed, and proteins contained within the vesicles.
To illustrate the above, we were requested to conduct qRT-PCR analysis on the isolated AIS-EVS. In the initial manuscript, we only presented the elevated expression of miR-30a-3p/5p, miR-30d-5p, and miR-30e-3p in hMSC following AIS-EVs treatment. Therefore, the expression of miR-30s in AIS-EVs was validated. We confirmed that hMSCs treated with AIS-EVs over-expressed these miR-30s after 24 hours of treatment compared to untreated hMSCs. Finally, we investigated whether the genes related to these miRNAs (SF4) were modulated by the treatment.
- In Figure 8C-D and Figure 9 we investigated the potential role of EVs from severe AIS-females in osteogenic differentiation. The effect of AIS-EVs was evaluated in comparison to the untreated hMSCs. Additionally, hMSCs were treated with EVs from healthy controls to demonstrate that vesicles from a healthy subject did not inhibit osteogenic differentiation. The effects on inhibition of osteogenic differentiation are linked to the particular content of the AIS vesicles (particular packaging of ncRNAs and proteins; the presence of miR-30 potentially affected osteogenesis). Further studies will allow us to look more closely at the effect of miR-30 family members regarding osteogenesis.
We realized that the meaning of the experiments was probably not well understood and added further explanations in the revised manuscript as mentioned above (Discussion section).
3) General. Authors added a serious number of references to their introduction and discussion parts. In particular, they highlight the studies by Wang et al (2020) and Chen et al (2022) who suggested circulating miR-151a-3p and miR-96-5p as potential biomarkers for severe AIS, respectively. How do the results of the present study compare with these publications? Do these 2 miRNAs (at least) behave similarly in terms of differential expression? Conversely, was the miR-30 family among previously published studies?
Both studies aimed to investigate the differentially expressed circulating miRNAs in AIS patients and HC. As we reported in the manuscript, Chen H et al. showed the over-expression of circulating miR-96-5p in plasma samples of AIS patients compared to HC. Wang et al, performed miRNA expression profiles in severe (n=5 females), mild (n=5 females) AIS patients and healthy controls (N=5 females) by miRNA sequencing. From their analysis they finally focused their attention on miR-151a-3p which resulted overexpressed in severe AIS patients, compared to HC.
Overall, data obtained confirmed the deregulation of miRNAs in AIS and their involvement in deregulation of bone metabolism. Interestingly, as commented by Wang, their results indicated that no specific miRNA was a good candidate for the moderate form of scoliosis. Rather, the authors hypothesised that: ''It is possible that deregulated miRNAs are exclusively involved in the progression of unbalanced spinal growth.'' We added this information in the Discussion section (The text is highlighted in yellow).
For us, it is interesting to note that they indicated miR-30c-5p and miR-30b-5p over-expressed in severe AIS vs HC. These results may confirm the importance of miR-30 family. We added this result in the discussion section (The text is highlighted in yellow).
In our study, we observed up-regulation of others miR-30 family members which were inserted in the MicroRNA Card we used (miR-30a-3p, miR-30a-5p, miR-30d-5p and miR-30e-3p). We also observed upregulation of miR-151-3p in AIS patients vs HC (Suppl Table 2 and 3). It is also important to note that our methodological approach has been different. Our preliminary results are encouring to deeper analyze miRNA in AIS patients through a new clinical protocol which will enroll a larger cohort of patients, establishing firstly the grade of severity during enrollment. We also want to better explore, in future studies, the possibility that miRNA deregulation in AIS is really associated with the form more severe and eventually try to explain the biological reason.
Overall, as we stressed in our study, the results obtained from all these studies are sometimes preliminary, with different methodological approaches but crucial for the experimental evidence obtained. In fact, they highlight the importance of deregulated miRNA expression in AIS pathogenesis and the possibility to validate biomarkers in the future.
4) Graphs in Fig 8 and 9. Why are the results of Fig 8B different in the revised manuscript compared to original submission (e.g., level of miR-30a-3p expression histogram)?
Also, why are statistical results different between the 2 versions? If authors changed their methods of statistical analysis, please explain why.
Homogenize the title size of graph axis in Fig 8A compared the other panels. We modified the title size.
We apologize for not commenting with the other reviewers on the changes requested by the third reviewer regarding the statistical analyses presented in the first version of the manuscript. We report below the comments/explanations that were provided following the request to improve/modify the statistics of the data presented, to implement the information in the Material and Methods section (also in the caption of the figures) and to use more rigorous statistical tests. Although the overall message remained unchanged, these modifications led to some adjustments in the analyses. However, EVs from severe AIS females negatively influenced osteogenesis and reduced the expression of factors involved in bone mineralization.
The answer given to Reviewer 3 in the first review is reproduced below to help clarify:
‘’ RESPONSE TO REVIEWER #3
13) Was data normally distributed?
Before performing inferential analysis for continuous data (qRT-PCR and ELISA data) the Shapiro Wilk test was applied.
14) Was the power of the study calculated?
The sample size was calculated using Bioconductor's 'size power' package (Sample Size and Power Calculation in Microarray Studies. v:1.40.0) considering a significance level α=0.05 and a power 1-β=0.90. The following parameters were considered for the sample size calculation: 1) hypothetical number of differentially expressed miRNAs equal to 8 out of 768 evaluated; 2) hypothetical number of false positives equal to 10% of the estimated; 3) effect size in miRNA expression equal to |log2(FC)| ≥ 1.5; 4) hypothetical standard deviation of the difference in miRNA expression between treatment and control equal to 0.98. The analysis estimated a minimum number of n=10 patients per group. However, due to the complexity and heterogeneity of AIS, such as the risk of disease exacerbation varying according to degrees of Cobb observed at diagnosis, gender, familiarity, etc., the number of AIS patients was doubled to ensure the power of the pilot study (Julien C et al. 2013).
15) We suggest that the authors work with a professional statistician to improve the reporting of data. For example, there are analyzes in the results that were not described in the methods and confidence intervals, not only p values should be reported. Graphs are out of scale.
We thank the reviewer for this important comment. We have carried out a new statistical analysis, which we have included in the Methods section. We have revised the text and captions to provide the necessary information.
16) Was ANOVA performed for the analysis in Figure 8?
We apologise for the mistake, but the data in Figures 8C and 8D were not analysed with ANOVA. We applied a one-way ANOVA followed by Dunnett's test for comparison with the control group. The statistics paragraph was implemented.’’
5) Figure 7. The title and values are missing/incomplete on the Y-axis of the right panel in Fig 7B.
We revised Figure 7B. The title and values are visible.
6) TLDA analysis. Authors used the term “microarray” when referring to the TLDA analysis, though this is a microfluidic qPCR platform. Please modify accordingly (lines 128 & 509). Also, the correct reference for TLDA cards is TaqMan™ Array Human MicroRNA A & B Cards v3.0 (line 748).
We have corrected the text.

Reviewer 3 Report
Comments and Suggestions for Authors
The authors have replied to our comments and the manuscript is now ready for publication.
Author Response
We appreciate the reviewer for reviewing and accepting our responses.